# Developing a class of dual atom materials for multifunctional catalytic reactions

Xingkun Wang[1,2,3,4], Liangliang Xu[5], Cheng Li[6,7,8], Canhui Zhang[1], Hanxu Yao[2,3,4], Ren Xu[1], Peixin Cui [9], Xusheng Zheng[10], Meng Gu [6,7], Jinwoo Lee [5] ✉, Heqing Jiang[2,3,4] ✉ & Minghua Huang [1] ✉

Dual atom catalysts, bridging single atom and metal/alloy nanoparticle catalysts, offer more opportunities to enhance the kinetics and multifunctional performance of oxygen reduction/evolution and hydrogen evolution reactions. However, the rational design of efficient multifunctional dual atom catalysts remains a blind area and is challenging. In this study, we achieved controllable regulation from Co nanoparticles to $CoN_4$ single atoms to $Co_2N_5$ dual atoms using an atomization and sintering strategy via an N-stripping and thermal-migrating process. More importantly, this strategy could be extended to the fabrication of 22 distinct dual atom catalysts. In particular, the $Co_2N_5$ dual atom with tailored spin states could achieve ideally balanced adsorption/desorption of intermediates, thus realizing superior multifunctional activity. In addition, it endows Zn-air batteries with long-term stability for 800 h, allows water splitting to continuously operate for 1000 h, and can enable solar-powered water splitting systems with uninterrupted large-scale hydrogen production throughout day and night. This universal and scalable strategy provides opportunities for the controlled design of efficient multifunctional dual atom catalysts in energy conversion technologies.

Vigorously developing the green hydrogen economy would bring great benefits for decarbonization in energy sectors[1–4]. Water splitting systems (WSSs), realized using renewable energy (e.g., solar energy) as the power and rechargeable batteries (e.g., metal-air batteries) as electricity storage, have been widely recognized as sustainable and $CO_2$-free energy devices for efficient and uninterrupted hydrogen ($H_2$) production[5–7]. Three core half-reactions of WSSs, namely the oxygen reduction reaction (ORR), oxygen evolution reaction (OER), and hydrogen evolution reaction (HER), are all subject to complicated multi-step proton-coupled electron transfer process, leaving the

challenging issues of sluggish kinetics and high overpotentials[8–14]. To date, the benchmark catalysts for improving these reaction efficiencies are precious-metal-based materials (i.e., $RuO_2/IrO_2$ only for OER, Pt/C only for ORR/HER), but their shortcomings, such as the high cost, scarcity, poor stability, and single-functionality, greatly restrict the large-scale application for series WSSs[15–18]. Although significant efforts toward these core half-reactions have been established in the widely acclaimed single atom catalyst (SAC) featuring minimum particle size, maximum atomic utilization, and well-defined metal-$N_4$ (M-$N_4$) active sites, their multifunctionality and sluggish ORR/OER/HER kinetics still

[1]School of Materials Science and Engineering, Ocean University of China, Qingdao, China. [2]Qingdao Key Laboratory of Functional Membrane Material and Membrane Technology, Qingdao Institute of Bioenergy and Bioprocess Technology, Chinese Academy of Sciences, Qingdao, China. [3]Shandong Energy Institute, Qingdao, China. [4]Qingdao New Energy Shandong Laboratory, Qingdao, China. [5]Department of Chemical and Biomolecular Engineering, Korea Advanced Institute of Science and Technology (KAIST), Yuseong-Gu, Daejeon, Republic of Korea. [6]Eastern Institute for Advanced Study, Eastern Institute of Technology, Ningbo, Zhejiang, PR China. [7]Department of Materials Science and Engineering, Southern University of Science and Technology, Shenzhen, China. [8]School of Physics and Astronomy, University of Birmingham, Birmingham, UK. [9]Key Laboratory of Soil Environment and Pollution Remediation, Institute of Soil Science, Chinese Academy of Sciences, Nanjing, China. [10]National Synchrotron Radiation Laboratory (NSRL), University of Science and Technology of China, Hefei, China. ✉e-mail: jwlee1@kaist.ac.kr; jianghq@qibebt.ac.cn; huangminghua@ouc.edu.cn

remain unsolved[19–24]. This arises from the obvious demerit of SAC, that is, only one kind of specific isolated active site, which makes it difficult to break the linear scaling relations between the adsorption/desorption toward complicated multi-intermediates and improve the catalytic ORR/OER/HER activities to the optimal level[25–27]. Dual atom catalysts (DACs) with adjacent dual atom sites have surfaced as a new research frontier that could bridge the SAC and metal/alloy nanoparticle catalysts, which endow DACs with integrated merits including high metal atom loading, more sophisticated and flexible active sites, easily modulated electronic structure, and the synergetic effects between two adjacent active sites[26,28–34]. Benefiting from these features, DACs offers more prospects toward conquering the challenges and limitations faced by SACs via synergistically adjusting the adsorption/desorption behaviors and activation of intermediates, thus accomplishing accelerated reaction kinetics and efficient multifunctional ORR/OER/HER performance[26,35,36]. Unfortunately, the rational design of highly efficient and robust DACs with multifunctionality is still in a blind area and with numerous challenges due to the lack of advanced fundamental knowledge of formation mechanisms for DACs.

In this study, a nanoparticle-to-single-atom-to-dual-atom (NP-to-SA-to-DA) atomization and sintering strategy was developed. We realized the controllable adjustment of the existing configuration states of Co species at the atomic level, yielding Co nanoparticles, $CoN_4$ single atoms, and $Co_2N_5$ dual atoms on N-doped hollow carbon spheres (termed $Co_{NP}$/HCS-900, $Co_{SA}$-N-HCS-900, and $Co_2$-N-HCS-900, respectively). This special design strategy allowed the investigation of the formation mechanism from NP to SA to DA. Density functional theory and experimental results demonstrated that Co atoms could be gradually stripped from the nanoparticles and trapped by N anchoring sites to form $CoN_4$ single atoms, and then spontaneously sintered into $Co_2N_5$ dual atoms via thermal migration. Moreover, it was found that the spin state of Co atoms can be harmonized from NP to SA to DA, in which the $Co_2N_5$ dual atom with a low spin state can achieve ideally balanced adsorption/desorption of O* and H* intermediates. As expected, the $Co_2$-N-HCS-900 affords the boosted multifunctional ORR/OER/HER activity, which endowed Zn-air batteries (ZABs) with excellent cycling charge–discharge stability over 800 h and enabled water splitting to operate for over 1000 h. The highly efficient and durable solar-powered WSS constructed using only $Co_2$-N-HCS-900 ensured uninterrupted large-scale $H_2$ production throughout the day and night for over 48 h. Most strikingly, this NP-to-SA-to-DA atomization and sintering strategy can be broadened to prepare 22 types of s-, p-, and d-block metal dual atom catalysts. This work provides a systematic study on the formation mechanisms and catalytic activities of nanoparticle, single atom, and dual atom catalysts, undoubtedly leading to an upsurge in the rational design of efficient and stable dual atom catalysts for applications in energy conversion technologies.

## Results and discussion

### Catalyst design concept

Density functional theory (DFT) calculations were conducted to reveal the structural transformation mechanism from nanoparticles to single atoms and then to dual atoms (NP-to-SA-to-DA) by taking $Co_{10}$ nanoparticles as examples[37,38]. First, the atomization process (Fig. 1a) was investigated from the $Co_{10}$ nanoparticles to two types of Co single atoms with different coordination of C (named $Co_{SA}$/C) and N atoms (named $CoN_4$). It can be seen that the formation of $Co_{SA}$/C from $Co_{10}$ nanoparticle decomposition requires overcoming a very high kinetic barrier of 3.23 eV with a large endothermicity of 3.22 eV, indicating that $Co_{10}$ nanoparticles might be the main form of existence. The formation of $CoN_4$ from $Co_{10}$ nanoparticles decomposition requires a relatively low kinetic barrier of 0.66 eV to be overcome but with a large exothermicity of 3.84 eV, manifesting that the introduction of N elements can promote the thermal atomization from $Co_{10}$ nanoparticles to $CoN_4$

single atoms. Additionally, the atomization processes of the $Co_{16}$ and $Co_4$ models were investigated. Figure S1 and S2 show that the formation of $Co_{SA}$/C from the decomposition of $Co_4$ and $Co_{16}$ nanoparticles requires overcoming a very high kinetic barrier of 3.97 and 3.65 eV, respectively, which is higher than that of the decomposition of $Co_{10}$ nanoparticles (3.23 eV). Moreover, the formation of $CoN_4$ from $Co_{10}$ nanoparticles decomposition requires a relatively low kinetic barrier (0.66 eV) to be overcome, lower than those for the formation of $CoN_4$ from the decomposition of $Co_4$ (1.50 eV) and $Co_{16}$ nanoparticles (1.30 eV). This implies that the transformation from $Co_{10}$ nanoparticles to single atoms can be achieved more easily. Figure 1b displays that the transforming processes, from two neighboring $CoN_4$ single atoms to edge-adjacent $Co_2N_6$ and then to the $Co_2N_5$ dual atom, were exothermic with the energy of 2.44 and 3.57 eV, respectively. Again, the processes from two randomly dispersed $CoN_4$ single atoms separated by several carbon atoms to edge-adjacent $Co_2N_6$ and then to $Co_2N_5$ dual atoms were exothermic (Figure S3), further confirming that randomly dispersed $CoN_4$ single atoms would spontaneously be sintered through the thermal migration process. Because the $Co_2N_5$ dual atom exhibited larger exothermicity than the $Co_2N_6$ dual atom, the former was more stable and dominantly formed during the sintering process. These results indicate that Co atoms could be gradually stripped from the $Co_{10}$ nanoparticles to form $CoN_4$ single atoms trapped by the anchoring sites of N, and then spontaneously sintered into $Co_2N_5$ dual atoms via thermal migration. Moreover, this NP-to-SA-to-DA atomization and sintering strategy can be broadened to the research scope of transforming conventional Al, Ca, Cr, Mn, Fe, Ni, Cu, Zn, Ru, Sb, Ce, Bi, and their alloys (taking the FeNi alloy as an example, Figure S4–S16).

Next, we investigated the electronic structures of $Co_{16}$ nanoparticle, $Co_{10}$ nanoparticle, $Co_4$ nanoparticle, $CoN_4$ single atom, and $Co_2N_5$ dual atom models (Figure S17). The charge density difference and Bader charge results shown in Fig. 1c confirm that the Co atoms of the $Co_2N_5$ dual atoms possess a higher charge of 1.11 e than those of the $CoN_4$ single atoms (1.03 e) and $Co_{10}$ nanoparticles (0.11 e). As shown in the projected density of states (PDOS) diagram (Figure S18), the electrons of Co-3d orbitals are asymmetrically arranged in the spin channels for $Co_{16}$ nanoparticles, $Co_{10}$ nanoparticles, $Co_4$ nanoparticles, $CoN_4$ single atoms, and $Co_2N_5$ dual atoms, exhibiting magnetic moments of 1.87, 1.48, 1.28, 0.81, and 0.05 $\mu_B$, respectively. The decreased spin magnetic moment from NP to SA to DA mainly results from the redistribution of electrons of the Co-3d orbital triggered by the energy level rearrangement, which gives rise to the increased filling degree of $d_{z^2}$ orbitals and induces the weakened adsorption of reaction intermediates (OOH*, O*, and OH* toward ORR/OER, H* toward HER), thus boosting the catalytic activities. Further support for this phenomenon can be demonstrated by the downshift of the d-band center of Co-3d orbitals from $Co_{16}$ nanoparticles to $Co_{10}$ nanoparticles to $Co_4$ nanoparticles to $CoN_4$ single atoms, to $Co_2N_5$ dual atoms, accompanied with the corresponding values of −0.52, −0.71, −1.27, −1.61, and −2.27 eV (Fig. 1d).

We also investigated the Gibbs free energies for the ORR/OER/HER of the $Co_{16}$ nanoparticle, $Co_{10}$ nanoparticle, $Co_4$ nanoparticle, $CoN_4$ single atom, and $Co_2N_5$ dual atom models to obtain a deeper understanding of the catalytic mechanism and origin of superior catalytic activities. The Gibbs free energy difference between $\Delta G_{OOH^*}$ and $\Delta G_{OH^*}$ ($|\Delta G_{OOH^*-OH^*}|$) can serve as an important ORR/OER reaction descriptor, with an ideal value of 2.46 eV[39–41]. The $|\Delta G_{OOH^*-OH^*}|$ is significantly restricted by the adsorption affinity of O*, in which either too strong or too weak adsorption of O* could result in the increased value of $|\Delta G_{OOH^*-OH^*}|$[42]. As shown in Fig. 1e, g, and S18, the spin magnetic moment presents a positive and quasi-linear correlation ($R^2 = 0.962$) with the $|\Delta G_{OOH^*-OH^*}|$, in the positive sequence from $Co_2N_5$ dual atoms (0.05 $\mu_B$, 2.55 eV) to $CoN_4$ single atoms (0.81 $\mu_B$, 3.12 eV) to $Co_4$ nanoparticles (1.28 $\mu_B$, 3.46 eV) to $Co_{10}$ nanoparticles (1.48 $\mu_B$, 3.79 eV) and to $Co_{16}$ nanoparticles (1.87 $\mu_B$, 4.36 eV). A universal $\Delta G_{OOH^*}$ to

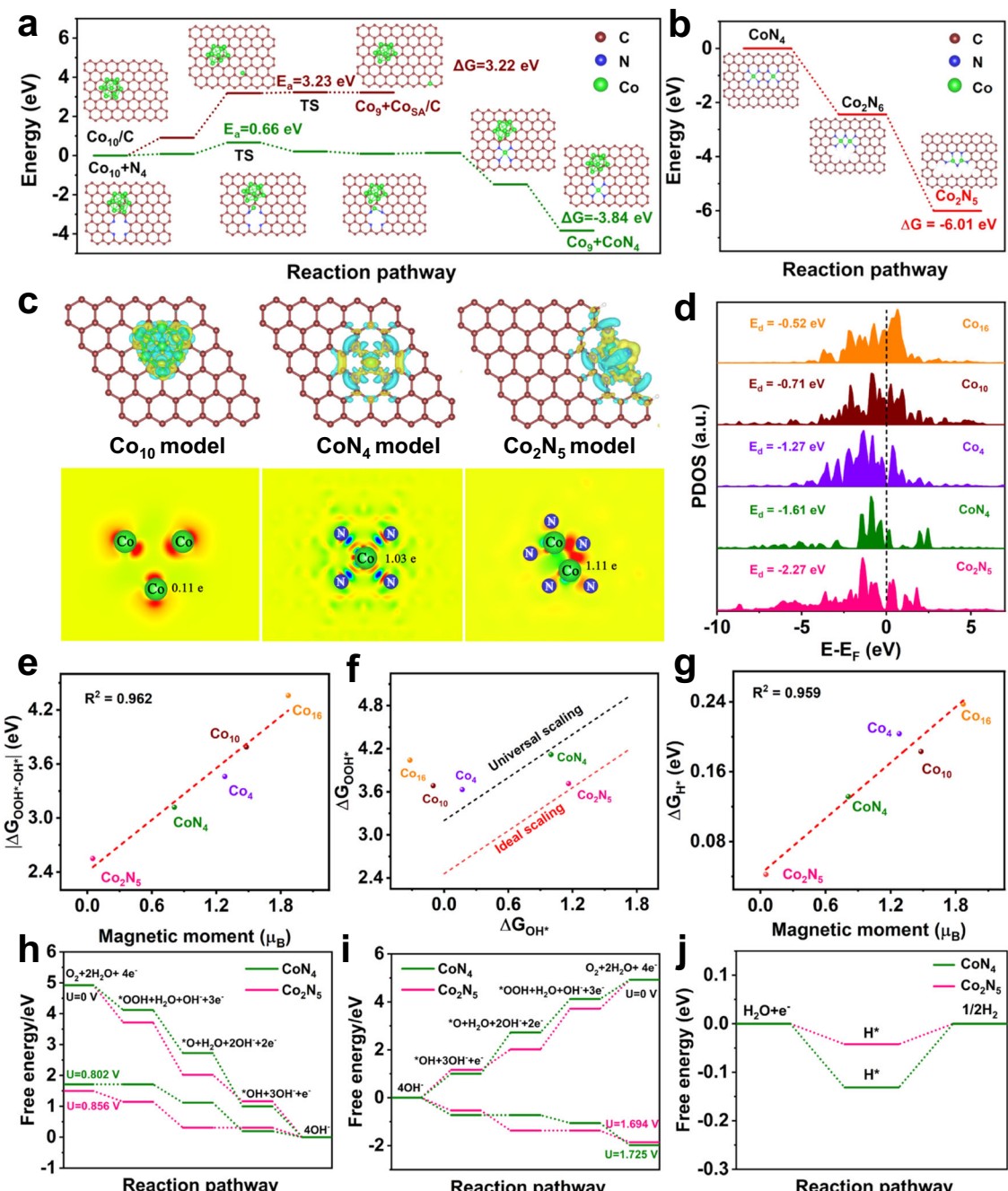

**Fig. 1 | DFT calculation. a** Calculated relative energies along the stretching pathway of the Co atom from $Co_{10}$ to $Co_{SA}/C$ or $CoN_4$ model by CI-NEB, **b** Calculated relative energies of $CoN_4$, $Co_2N_6$, and $Co_2N_5$ models. **c** Charge density difference and Bader charge diagrams; **d** PDOS; **e** linear correlation between magnetic moment and $|\Delta G_{OOH^*-OH^*}|$; **f** the $\Delta G_{OOH^*}$ to $\Delta G_{OH^*}$ scaling for $Co_{16}$, $Co_{10}$, $Co_4$, $CoN_4$, and $Co_2N_5$ models relative to the universal and ideal scaling lines; and **g** linear correlation between the magnetic moment and $\Delta G_{H^*}$. Free energy diagrams of $CoN_4$ and $Co_2N_5$ models for **h** ORR, **i** OER, and **j** HER.

$\Delta G_{OH^*}$ scaling relation with an average $|\Delta G_{OOH^*-OH^*}|$ value of 3.2 eV has been established for most conventional catalysts, whereas the ideal $\Delta G_{OOH^*}$ to $\Delta G_{OH^*}$ scaling relation possesses an average $|\Delta G_{OOH^*-OH^*}|$ value of 2.46 eV for ideal catalysts (Fig. 1f)[39,40]. The $\Delta G_{OOH^*}$ to $\Delta G_{OH^*}$ coordinate point of the $Co_2N_5$ dual atom is located at the ideal scaling relations accompanied with the $|\Delta G_{OOH^*-OH^*}|$ value of 2.55 eV, which is very close to the 2.46 eV for ideal catalysts toward ORR/OER. This indicates that the energy of O* intermediate adsorption/desorption for $Co_2N_5$ was the most appropriate. The possible reason for this phenomenon is that the $Co_2N_5$ dual atom with the decreased spin magnetic moment can break the universal $\Delta G_{OOH^*}$ to $\Delta G_{OH^*}$ scaling relation, thus achieving the ideal balanced O* adsorption. This phenomenon is

further supported by the moderate O* adsorption energy (1.89 eV, Figure S19) of the $Co_2N_5$ dual atom model among three investigated models, indicating that it could achieve optimized O* adsorption/desorption, thus boosting the ORR/OER activities. To determine the correlation between the spin configuration and the free energy of O*, the crystal orbital Hamilton population (COHP) was calculated to compare the bonding character of O* absorbed onto the $Co_{10}$, $CoN_4$, and $Co_2N_5$ models. Positive and negative COHP are due to bonding and antibonding states, respectively. The Co-O bonding strength can be evaluated using the integrated COHP (ICOHP) values, which could quantitatively describe the *d-p* hybridization strength. As depicted in Figure S20, the ICOHP values were found to be −0.22 for $Co_{10}$, −0.47

for $CoN_4$, and −0.45 for $Co_2N_5$. The intermediate ICOHP value of the three investigated models confirms the moderated Co-O affinity in the $Co_2N_5$ model, suggesting optimal O* adsorption/desorption, enhancing the ORR/OER activities. In Figure S21, a pronounced antibonding state for the $Co_2N_5$ model emerges at the Fermi level compared to the other models, implying greater electron transfer from the Co-3$d$ orbital to the vacant O-2$p$ orbital. This leads to reduced reaction activation energy and improved catalyst conductivity with minimal ohmic loss, enhancing catalytic activity[43,44]. Regarding HER, a negative and quasi-linear correlation exists between the spin magnetic moment and the Gibbs free energy of H* ($\Delta G_{H*}$). Specifically, $\Delta G_{H*}$ decreases linearly ($R^2 = 0.959$) with the increased spin magnetic moment (Fig. 1g). Among the studied models, the $Co_2N_5$ dual atom possesses the highest $\Delta G_{H*}$ of −0.04 eV (close to the ideal value of 0 eV), demonstrating that its low spin state facilitates the moderate adsorption and desorption of the H* intermediate. Moreover, the highest H* adsorption energy of −0.21 eV was obtained among the three investigated models (Figure S22), again validating that it affords moderate adsorption and desorption of H*, endowing excellent HER activities.

Figure 1h–j and S23–27 depict the optimized atomic configurations and Gibbs free energies of the $Co_{16}$ nanoparticle, $Co_{10}$ nanoparticle, $Co_4$ nanoparticle, $CoN_4$ single atom, and $Co_2N_5$ dual atom models bonded with the adsorption of reaction intermediates for catalyzing the ORR/OER/HER. For ORR (Fig. 1h and S25), the $Co_2N_5$ dual atom affords a higher thermodynamic limiting potential of 0.856 V than both the $CoN_4$ single atom (0.802 V) and $Co_{10}$ nanoparticles (−0.102 V), highlighting its good ORR activity. As shown in Fig. 1i and S26, the lowest thermodynamic limiting potential of 1.694 V toward the OER was obtained for the $Co_2N_5$ dual atom among the investigated models, indicating that it shows the lowest overpotential and superior OER efficiency. Figure 1j and S27 show the Gibbs free energy of H* intermediates toward the HER across the models, where the $Co_2N_5$ dual atom provides an adsorption energy value of H* of −0.04 eV, closest to 0 eV, indicating its superior HER performance. Based on DFT analyses, the $Co_2N_5$ dual atom has a tailored spin state that promotes the moderated and balanced adsorption and desorption of reaction intermediates toward the ORR/OER/HER, enhancing its trifunctional performance[45,46].

## Synthesis and structural characterization

Driven by DFT analysis, a series of catalysts were prepared, including $Co_{NP}$/HCS-900 with aggregated Co nanoparticles, $Co_{SA}$-N-HCS-900 with $CoN_4$ single atoms, and $Co_2$-N-HCS-900 with paired $Co_2N_5$ dual atoms (Fig. 2a). A facile double-solvent impregnation method was used to prepare the Co-hollow polymer spheres (Co-HPS) precursor. Then, $Co_{NP}$/HCS-900 was obtained by directly pyrolyzing the Co-HPS, and the $Co_{NP}$/N-HCS-300 were synthesized by pyrolyzing the Co-HPS with melamine at 300 °C. Further, $Co_{SA}$-N-HCS-900 can be synthesized when the melamine was added in the above pyrolysis process since the melamine can be decomposed at high temperatures (>400 °C) and serve as the N agent that is generally coordinated with Co atoms to form Co-Nx moieties for promoting the Co nanoparticles atomization[47]. By effectively revising the calcination time, the single Co atoms could couple with each other and sinter to form $Co_2$-N-HCS-900 with the paired $Co_2N_5$ dual atom dimer via thermal migration. The X-ray diffraction (XRD) patterns in Figure S28 show that only $Co_{NP}$/HCS-900 has a diffraction peak indexing crystalline Co nanoparticle, whereas no peak can be observed for either $Co_{SA}$-N-HCS-900 or $Co_2$-N-HCS-900. Further support for this phenomenon can be provided by their corresponding transmission electron microscope (TEM) and high-resolution TEM (HRTEM) images. As shown in Figure S29, numerous nanoparticles, accompanied by a lattice distance of 0.203 nm, indexing the (111) facet of metallic Co, were uniformly anchored on the hollow carbon spheres of $Co_{NP}$/HCS-900. No obvious Co nanoparticles are observed in either $Co_{SA}$-N-HCS-900 (Fig. 2b–i) or

$Co_2$-N-HCS-900 (Fig. 2j–q). Moreover, both $Co_{SA}$-N-HCS-900 and $Co_2$-N-HCS-900 exhibited ring-like diffraction in the selected area electron diffraction patterns (Figure S30 and S31) and a uniform distribution of C, N, O, and Co elements in the associated elemental mapping profiles (Figs. 2d–e, l–m), indicating the atomic dispersion of Co species on both $Co_{SA}$-N-HCS-900 and $Co_2$-N-HCS-900.

Aberration-corrected high-angle annular dark-field scanning transmission electron microscopy (AC HAADF-STEM) was used to investigate the atomic states of the Co species in $Co_{NP}$/N-HCS-300, $Co_{SA}$-N-HCS-900, and $Co_2$-N-HCS-900. As shown in Figure S32, several aggregated nanoparticles (marked with red cycles) were captured on the carbon substrate, indicating that some Co nanoparticles had surfaced on $Co_{NP}$/N-HCS-300. The AC HAADF-STEM images of $Co_{SA}$-N-HCS-900 (Fig. 2f–g) and $Co_2$-N-HCS-900 (Fig. 2n–o) displays numerous bright dots (marked with red circles) homogeneously dispersed on the carbon skeleton, which indicate Co atoms due to the difference in Z-contrast between the heavier Co and the lighter C and N. The projected distance between two adjacent Co atoms in $Co_{SA}$-N-HCS-900 is mainly distributed in the range of 0.30–0.50 nm (Fig. 2h–i and S33), indicating that the Co species mainly exist as single atoms. As for $Co_2$-N-HCS-900 (Fig. 2o), a large proportion of Co atoms are adjacent to each other and presented in the form of dual Co atoms, with the Co-Co distance ranging from 0.12 nm to 0.25 nm (Fig. 2p–q and S34). This verifies the presence of paired $Co_2$ dual atom dimers. Figure S35 reveals numerous pores (marked by green circles) surrounding the paired $Co_2$ dual atom dimers, indicating that the dimers might be positioned at the edge of the carbon framework[28,48,49]. This suggests that by adjusting calcination durations, single Co atoms can merge to create edge-adjacent paired $Co_2$ dual atom dimers. Additionally, this NP-to-SA-to-DA atomization and sintering strategy can be adopted for the generalized synthesis of 21 types of edge-adjacent paired $s$-, $p$-, and $d$-block $M_2$ dual atom structures, including edge-adjacent $Al_2$, $Ca_2$, $Cr_2$, $Mn_2$, $Fe_2$, $Ni_2$, $Cu_2$, $Zn_2$, $Ru_2$, $Sb_2$, $Ce_2$, $Bi_2$, CoFe, CoNi, CoCu, CoZn, CoMn, FeNi, FeCu, FeZn, and FeMn dual atoms on N-HCS, which were confirmed by the AC HAADF-STEM and the corresponding 3D atom-overlapping Gaussian-function fitting map analysis (Fig. 3 and S36).

## Atomic coordination structure analysis

X-ray absorption spectroscopy (XAS) was employed to elucidate the detailed local structures of the Co atoms in $Co_{SA}$-N-HCS-900 and $Co_2$-N-HCS-900. As can be seen from the X-ray absorption near edge structure (XANES) spectra in Fig. 4a, the Co K-edge XANES curves for $Co_2$-N-HCS-900 and $Co_{SA}$-N-HCS-900 lie between those for Co foil and $Co_3O_4$, indicating that their Co atoms show a positive valence. Figure S37 displays the Fourier transform (FT) $k^3$-weighted extended X-ray absorption fine structure (FT-EXAFS) spectra at R spaces, in which the main peak at 1.43 Å that corresponds to the Co-N shell (relative to CoPc) is obtained for both $Co_{SA}$-N-HCS-900 and $Co_2$-N-HCS-900. The secondary peak appearing at R = 2.25 Å was observed for $Co_2$-N-HCS-900, which can be indexed to the Co-Co shell in reference to Co foil, indicating the formation of Co-Co bonds in the catalysts. In addition, a comparison of the q-space magnitudes in Fig. 4b confirms the existence of Co-N and Co-Co paths in $Co_2$-N-HCS-900, whereas only the Co-N path exists in $Co_{SA}$-N-HCS-900. Further support for the Co-Co path in $Co_2$-N-HCS-900 is provided by wavelet transform (WT) contour plots (Fig. 4c). To determine the detailed coordination structure of the Co atoms in both $Co_{SA}$-N-HCS-900 and $Co_2$-N-HCS-900, theoretical calculations and fitted EXAFS at the Co K-edge were conducted. As shown in Fig. 4d and Table S1, $Co_{SA}$-N-HCS-900 provides an average Co-N/O coordination number of 4.0 and a Co-Co coordination number of 0.1, indicating that Co atoms predominantly exist in the form of a $CoN_4$ structure (inset of Fig. 4d). As for $Co_2$-N-HCS-900, the average coordinated number of Co-N/O bond and Co-Co bond were determined to be 3.9 and 0.8 (close to 1), respectively, meaning that most Co atoms are preferentially bonded with one Co

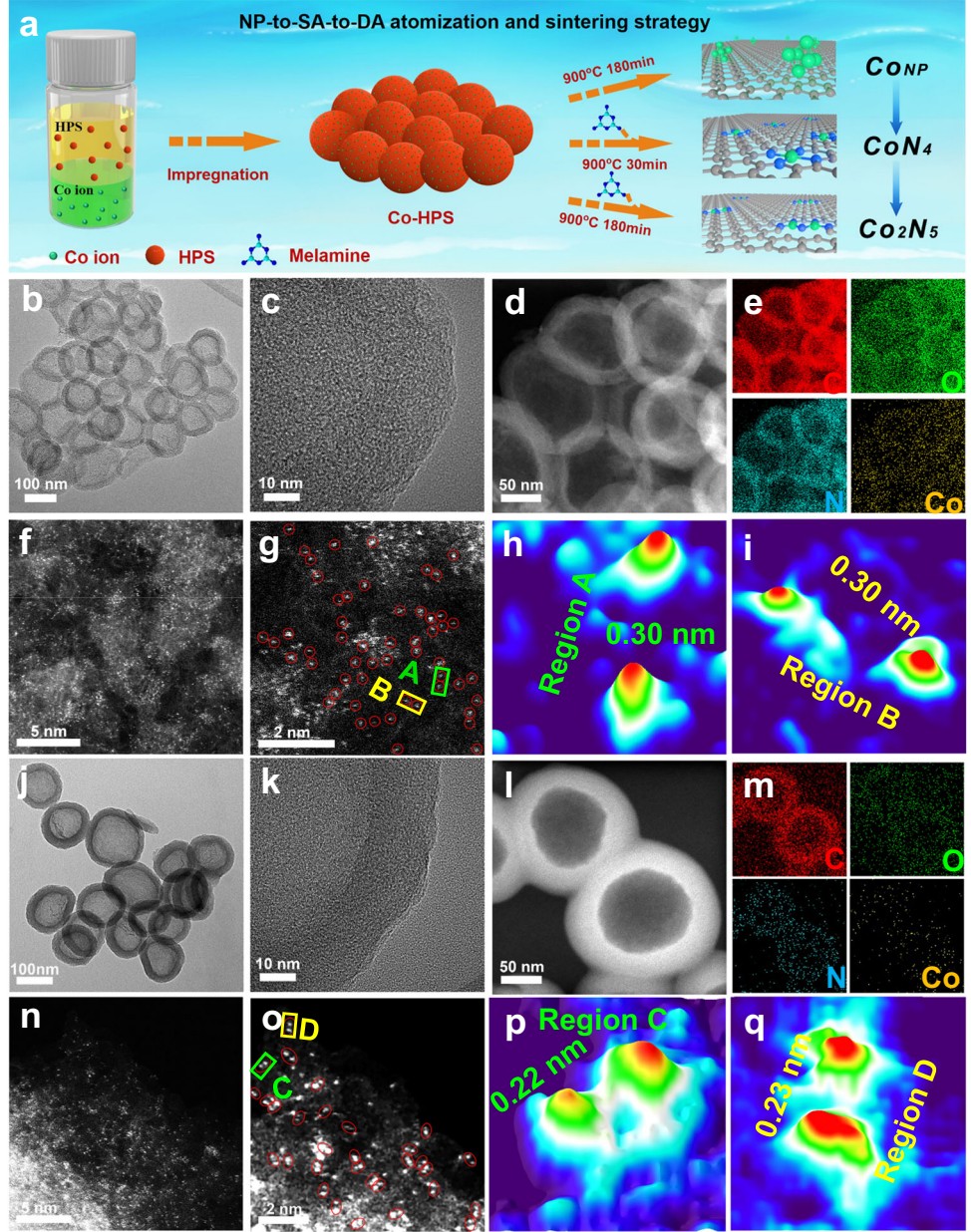

**Fig. 2 | Structural characterizations. a** Schematic diagram of the NP-to-SA-to-DA atomization and sintering strategy. **b** TEM, **c** HRTEM, **d** HAADF-STEM, and **e** C, O, N, Co elemental mapping images; **f**, **g** AC HAADF-STEM images (Co single atoms marked by red circles); **h**, **i** 3D atom-overlapping Gaussian-function fitting map in **g** for Co$_{SA}$-N-HCS-900. **j** TEM, **k** HRTEM, **l** HAADF-STEM, and **m** C, O, N, Co elemental mapping images; **n**, **o** AC HAADF-STEM images (Co dual atoms marked by red circles); **p**, **q** 3D atom-overlapping Gaussian-function fitting map in **o** for Co$_2$-N-HCS-900.

and three N atoms (and one O) to form Co$_2$N$_5$-O structure (inset of Fig. 4e). To further verify the possible structures of Co$_{SA}$-N-HCS-900 and Co$_2$-N-HCS-900, DFT calculations were performed to investigate the possible structures containing a single Co atom and a paired Co$_2$ structure (models 1 to 10, Fig. S38). Moreover, a comparison between the simulated EXAFS and XANES spectra of the possible structures and the experimental spectra was recorded. As shown in Figure S39 and S40 and Table S2, the simulated spectra based on the single atom CoN$_4$ model (model 9) agreed well with the experimental EXAFS and XANES results of Co$_{SA}$-N-HCS-900, confirming that this model is the most likely actual structure of Co$_{SA}$-N-HCS-900. For Co$_2$-N-HCS-900, the simulated spectra based on model 2 (binuclear Co$_2$N$_5$ configurations with oxygen) agreed well with the experimental EXAFS and XANES results (Figure S41 and S42 and Table S3), indicating that model 2 is the most likely actual structure. These results synergistically validated that

CoN$_4$ single atoms could couple with each other to form paired Co$_2$N$_5$ dual atom dimers via thermal migration.

X-ray photoelectron spectroscopy (XPS) was conducted to explore the chemical elements and states of Co$_{NP}$/HCS-900, Co$_{SA}$-N-HCS-900, and Co$_2$-N-HCS-900 (Figure S43a). The high-resolution C 1$s$ XPS spectra of the three investigated catalysts (Figure S44) can be divided into four peaks at approximately 284.7, 285.2, 286.3, and 290.3 eV, which are indexed to $sp^2$ C = C and $sp^3$ C-C (defects), C-N, and COOH species, respectively. The high-resolution N 1$s$ XPS spectra in Figure S45 confirm the coexistence of pyridinic N, Co-Nx, graphitic N, and oxidized N[50]. Figure S46 shows that Co$_{SA}$-N-HCS-900 affords a higher content of C-N, pyridinic N, and Co-Nx species than Co$_{NP}$/HCS-900, indicating that melamine (N agent) could be coordinated with Co atoms to form Co-Nx moieties to promote Co nanoparticle atomization. Further, both the content of pyridinic N and C-N species show a

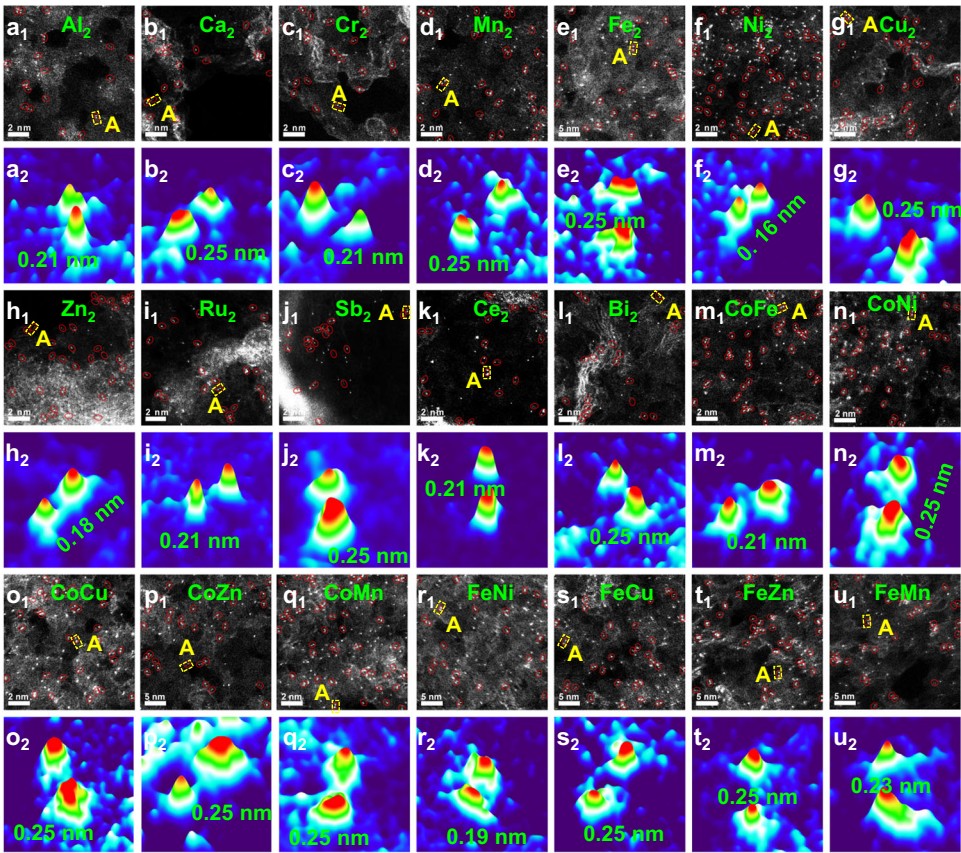

**Fig. 3 | Structural characterizations of 21 types of s-, p-, and d-block dual atom catalysts.** AC HAADF-STEM and corresponding 3D atom-overlapping Gaussian-function fitting map of region A for **a** Al₂-N-HCS-900, **b** Ca₂-N-HCS-900, **c** Cr₂-N-HCS-900, **d** Mn₂-N-HCS-900, **e** Fe₂-N-HCS-900, **f** Ni₂-N-HCS-900, **g** Cu₂-N-HCS-900, **h** Zn₂-N-HCS-900, **i** Ru₂-N-HCS-900, **j** Sb₂-N-HCS-900, **k** Ce₂-N-HCS-900, **l** Bi₂-N-HCS-900, **m** CoFe-N-HCS-900, **n** CoNi-N-HCS-900, **o** CoCu-N-HCS-900, **p** CoZn-N-HCS-900, **q** CoMn-N-HCS-900, **r** FeNi-N-HCS-900, **s** FeCu-N-HCS-900, **t** FeZn-N-HCS-900, and **u** FeMn-N-HCS-900.

downward trend with calcination time, revealing that the C-N bonds coordinated with pyridinic N are preferentially cleaved[51,52]. This would lead to a decrease in total N content (Figure S43b) and generate many defective structures, which is further evidenced by the high $I_D/I_G$ ratio (1.48, Figure S47) and large Brunauer–Emmett–Teller (BET) surface area (772.9 $m^2\,g^{-1}$, Figure S48a) and pore volumes (1.13 $cm^3\,g^{-1}$, Figure S48b) for Co₂-N-HCS-900[53–55]. The Co 2$p$ XPS spectra in Figure S49 shows that four peaks attributed Co-N species and corresponding satellite peaks are obtained for the investigated catalysts, while two peaks indexing Co⁰ species are observed in the Co_NP/HCS-900, indicating the existence of metallic Co nanoparticles in Co_NP/HCS-900. As displayed in Figure S50, the deconvolution of the O 1$s$ spectra demonstrated the coexistence of oxygen-containing functional groups (C = O at ca. 531.8 eV, COOH at ca. 533.3 eV, and absorbed water at ca. 536.3 eV) in Co_NP/HCS-900, Co_SA-N-HCS-900, and Co₂-N-HCS-900. Interestingly, a new peak appears at 530.2 eV that was indexed to the Co-O bond for Co₂-N-HCS-900, indicating the presence of Co-O bonds in the catalyst. The Co content, measured by inductively coupled plasma optical emission spectrometry (ICP-OES), was determined to be 1.48 wt% for Co_NP/HCS-900, 1.41 wt% for the Co_SA-N-HCS-900, and 1.74 wt% for the Co₂-N-HCS-900. Considering the BET surface area and Co content, the Co₂-N-HCS-900 affords a higher atomic Co coverage of 0.233 atoms per square nanometer than the Co_SA-N-HCS-900 (0.196 atoms per square nanometer), demonstrating more accessible active Co sites on the former.

X-band electron paramagnetic resonance (EPR) measurement is a powerful tool to investigate the paramagnetic properties of catalysts. As shown in Fig. 4f, a strong signal was detected for Co_NP/HCS-900, indicating that it is paramagnetic. A downward trend was observed for Co_NP/HCS-900 to Co_SA-N-HCS-900 to Co₂-N-HCS-900, indicating that the spin magnetic moment decreased from the Co nanoparticles to single CoN₄ sites and then to paired Co₂N₅ sites. Notably, no signal appeared for Co₂-N-HCS-900, possibly because of the formation of a binuclear Co structure with antiferromagnetic coupling sites, again confirming that the paired Co₂N₅ structure was successfully constructed in Co₂-N-HCS-900. As shown in Fig. 4g, the ferromagnetic hysteresis loops of the investigated catalysts at 300 K exhibited saturation magnetization. Notably, the saturation magnetization decreased from Co_NP/HCS-900 to Co_SA-N-HCS-900 to Co₂-N-HCS-900. An enlarged view of the curve around H = 0 indicates that the Co₂-N-HCS-900 provides the lowest coercive magnetic field and residual magnetization (Fig. 4g). To further reveal the electron-spin configurations of the investigated catalysts, zero-field cooling (ZFC) temperature-dependent magnetic susceptibility measurements were conducted (Fig. 4h and S51). As presented in Fig. 4h and i, the effective magnetic moment of Co_SA-N-HCS-900 and Co₂-N-HCS-900 were calculated to be 2.6 and 1.7 $\mu_{eff}$, respectively. The average number of unpaired electrons is 1.0 in the Co-3$d$ orbitals for Co₂-N-HCS-900, which is lower than that of Co_SA-N-HCS-900 (1.7), indicating decreased electron spin polarization from Co_SA-N-HCS-900 to Co₂-N-HCS-900. Based on these results, it can be concluded that the spin magnetic moment exhibits a downward trend from Co_NP/HCS-900 to Co_SA-N-HCS-900 to Co₂-N-HCS-900. Ultraviolet photoemission spectroscopy (UPS) was used to investigate the electronic states of Co_NP/HCS-900, Co_SA-N-HCS-900, and Co₂-N-HCS-900. The work function represents the minimum energy required for the inner electrons to escape from the catalyst surface. As shown in Fig. 4j and S52, the work function of Co₂-N-HCS-900 was determined to be 5.04 eV, which is higher than

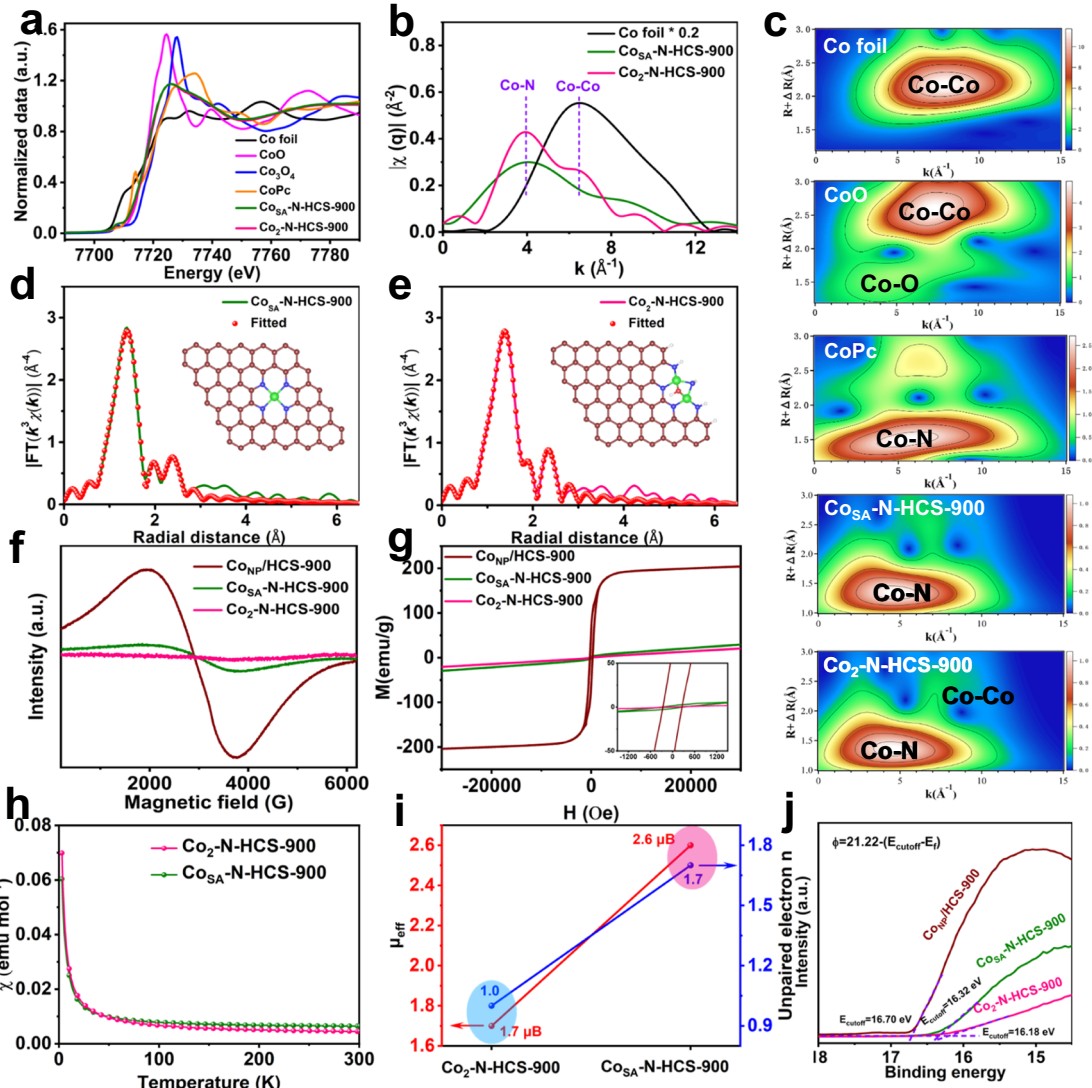

**Fig. 4 | Local structural characterizations and electron spin-state analysis.**
**a** XANES spectra, **b** q-space magnitude comparisons, and **c** WT-EXAFS at Co K-edge of Co foil, CoO, CoPc, $Co_{SA}$-N-HCS-900, and $Co_2$-N-HCS-900; **d**, **e** $k^3$-Weighted EXAFS fitting curves at R space for $Co_{SA}$-N-HCS-900 and $Co_2$-N-HCS-900 (inset: schematic structure of $CoN_4$ and $Co_2N_5$ models; green: Co, blue: N, red: O); **f** X-band electron paramagnetic resonance (EPR) spectra and **g** magnetic hysteresis loops at room temperature (300 K) and inset image of the magnification of magnetic hysteresis loops around H = 0; **h** M-T curves and **i** corresponding unpaired electron $n$ and effective magnetic moment ($\mu_{eff}$) for $Co_{SA}$-N-HCS-900 and $Co_2$-N-HCS-900. **j** Work function spectra for $Co_{NP}$/HCS-900, $Co_{SA}$-N-HCS-900, and $Co_2$-N-HCS-900.

those of $Co_{SA}$-N-HCS-900 (4.90 eV) and $Co_{NP}$/HCS-900 (4.52 eV). The valence band maximum is referred to as the highest occupied molecular orbital (HOMO) and is related to the highest energy level of the valence band in the solid material. It is widely recognized that shifts in the valance band indicate changes in the $E_d$ energy level, primarily because valence electrons near the Fermi level significantly influence the $d$ states[56]. The $Co_2$-N-HCS-900 exhibits a high calculated valance band maximum (VBM) value of 5.63 eV compared to that of $Co_{SA}$-N-HCS-900 (5.00 eV) and $Co_{NP}$/HCS-900 (4.41 eV) (Fig. 4j, S52, and S53), indicating that the valance band gets away from the Fermi level for the $Co_2$-N-HCS-900. The larger work function and VBM indicate that $Co_2$-N-HCS-900 presents a higher energy barrier for electron donation and possesses a reduced $E_d$ energy level, resulting in favorable interactions between the intermediates and active sites and enhanced reaction activity[56].

## Evaluation of electrochemical performance
The ORR/OER/HER performances of the obtained catalysts were then investigated (Fig. 5a–f). Linear sweep voltammetry (LSV) curves show

that $Co_2$-N-HCS-900 exhibits a more positive onset potential (0.99 V) and half-wave potential (0.86 V) toward ORR than HCS-900, $Co_{NP}$/HCS-900, and $Co_{SA}$-N-HCS-900 (Fig. 5a and S54), which are comparable to those of commercial Pt/C and other reported non-precious M-N-C catalysts (Table S4). The superior ORR kinetics of $Co_2$-N-HCS-900 are evident as it demonstrates the smallest Tafel slope (48.0 mV dec$^{-1}$) and highest kinetic current density (8.33 mA cm$^{-2}$ @ 0.85 V), as shown in Fig. 5d and S55. Moreover, $Co_2$-N-HCS-900 exhibits a higher electrochemical double-layer capacitance ($C_{dl}$) value of 195.1 mF cm$^{-2}$ and mass activity of 97.6 A g$^{-1}_{Co}$ @ 0.9 V than the control catalysts ($Co_{NP}$/HCS-900, $Co_{SA}$-N-HCS-900, and commercial Pt/C). This suggests that more abundant and accessible active sites for ORR catalysis are present (Figure S56–58). Koutecky–Levich (K–L) plots derived from the LSV curves indicate that the ORR kinetics of $Co_2$-N-HCS-900 are closely related to the $O_2$-diffusion process (Figure S59). The electron transfer number ($n$) is determined to be 3.66, indicating a direct $4e^-$ reduction pathway during ORR (Figure S59). Additionally, $Co_2$-N-HCS-900 exhibits the lowest OER overpotentials of 333 mV at 10 mA cm$^{-2}$ and smallest Tafel slope (97.1 mV dec$^{-1}$) of the investigated catalysts,

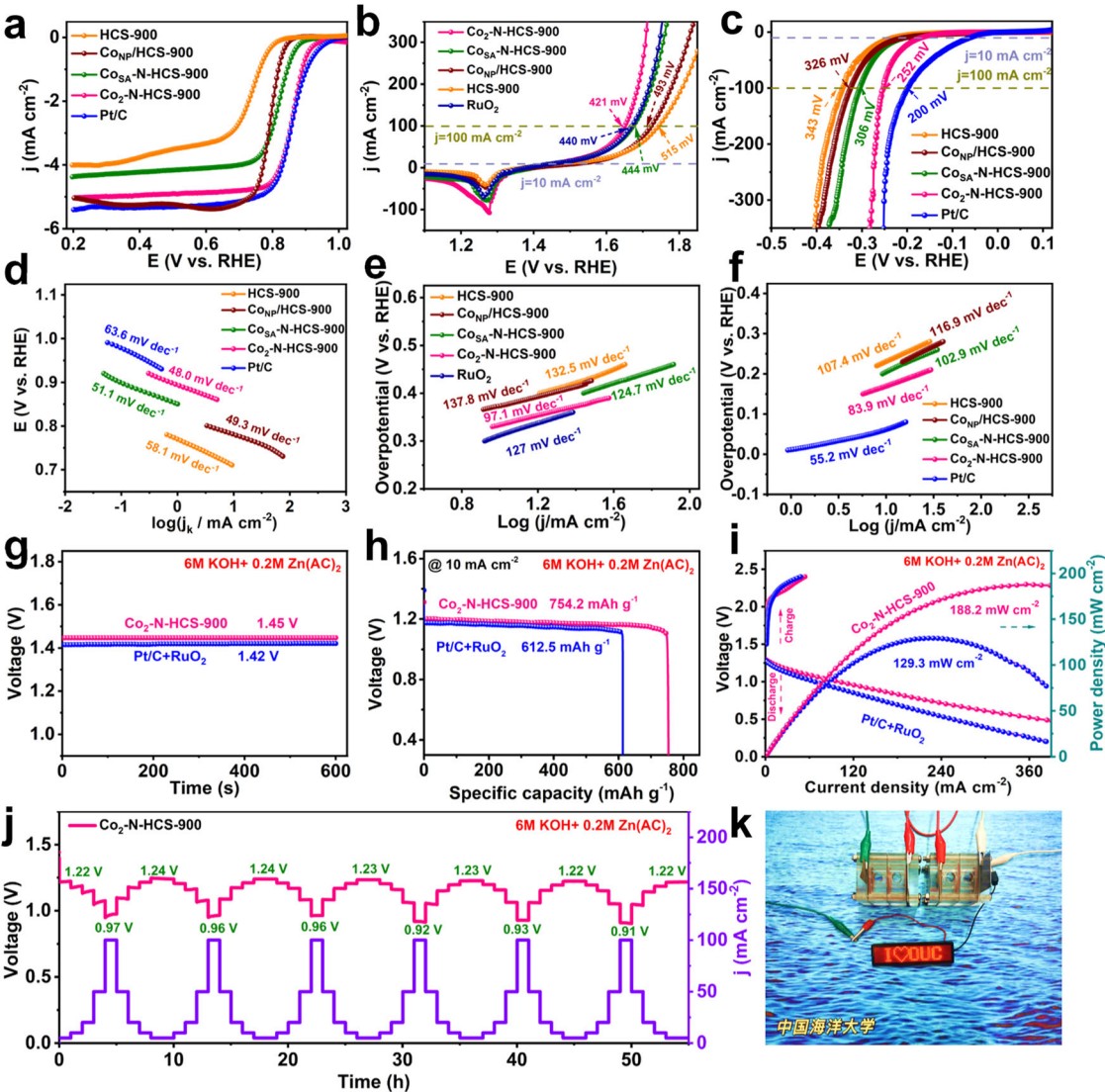

**Fig. 5 | Electrocatalytic activity and the performance of ZABs. a** LSV curves and **d** Tafel plots toward ORR in 0.1 M KOH; **b** LSV curves and **e** Tafel plots toward OER; and **c** LSV curves and **f** Tafel plots toward HER in 1.0 M KOH for all investigated and commercial (Pt/C or RuO₂) catalysts. **g** Open-circuit potential plots; **h** specific capacities; and **i** discharge polarization, charge polarization, and corresponding power density curves for ZABs driven by Co₂-N-HCS-900 or commercial Pt/ C + RuO₂. **j** Discharge curves of ZABs at various discharge current densities, and **k** LED screen powered by two tandem Co₂-N-HCS-900-based ZABs.

rivaling that of commercial RuO₂ and other advanced M-N-C catalysts (Figs. 5b, e, and Table S5). For the HER, Co₂-N-HCS-900 achieved good activity in view of its low overpotential (166 mV at 10 mA cm⁻² and 252 mV at 100 mA cm⁻²) and small Tafel slopes (83.9 mV dec⁻¹), surpassing those of HCS-900, Co$_{NP}$/HCS-900, and Co$_{SA}$-N-HCS-900 and comparable to those of most reported M-N-C catalysts (Table S6). Further, Co₂-N-HCS-900 exhibits a large mass activity of 2.79 A g⁻¹$_{Co}$ @ η = 400 mV toward OER and 1.31 A g⁻¹$_{Co}$ @ η = 200 mV toward HER, exceeding those of the control Co$_{NP}$/HCS-900, Co$_{SA}$-N-HCS-900, and commercial Pt/C (Figure S60 and S61). These results highlight the significant role of the adjacent Co atoms in the Co₂N₅ structure in realizing advanced trifunctional activities, which also have advantages over currently reported multifunctional single atom catalysts (Table S7). Moreover, Co₂-N-HCS-900 confers superior long-term ORR/OER/HER stability compared to commercial catalysts (Pt/C or RuO₂, Figure S62–64).

The ORR/OER activity of Co₂-N-HCS-900 motivated us to investigate its practical application in assembled liquid ZABs. Figure 5g–i and Table S8 show that the ZABs driven by Co₂-N-HCS-900 demonstrate a high open-circuit potential (OCP) of 1.45 V, a large specific capacity of 754.2 mAh g⁻¹, and an eminent peak power density of 188.2 mW cm⁻², far surpassing commercial Pt/C + RuO₂ based ZABs (1.42 V, 612.5 mAh g⁻¹, and 129.3 mW cm⁻²). The discharge curves in Fig. 5j show that Co₂-N-HCS-900-based ZABs provide a voltage of 1.22 V (5 mA cm⁻²) and 0.97 V (100 mA cm⁻²) during the first cycle, and the voltage loss is negligible after 6 cycles, validating its excellent rate performance and reversibility. The ZABs powered by Co₂-N-HCS-900 also possessed an excellent round-trip efficiency of 58.1% at a current density of 5 mA cm⁻² and an ultralong lifespan of over 800 h with negligible round-trip efficiency fading (Figure S65 and S66). Notably, the Co₂-N-HCS-900-based ZABs can operate over 600 cycles under a high current density of 50 mA cm⁻² (Figure S67), demonstrating its promising practical application. Moreover, two or three tandem Co₂-N-HCS-900-based ZABs offer an OCP of 2.91 or 4.35 V, respectively, which could power a light-emitting diode screen (LED, 2 V) for several hours (Fig. 5k and S68).

Leveraging its good bifunctional OER/HER activity, water electrolysis devices were assembled using Co₂-N-HCS-900 as the anode and cathode catalysts. The Co₂-N-HCS-900-based system requires potentials of approximately 1.76, 1.97, and 2.02 V to achieve current

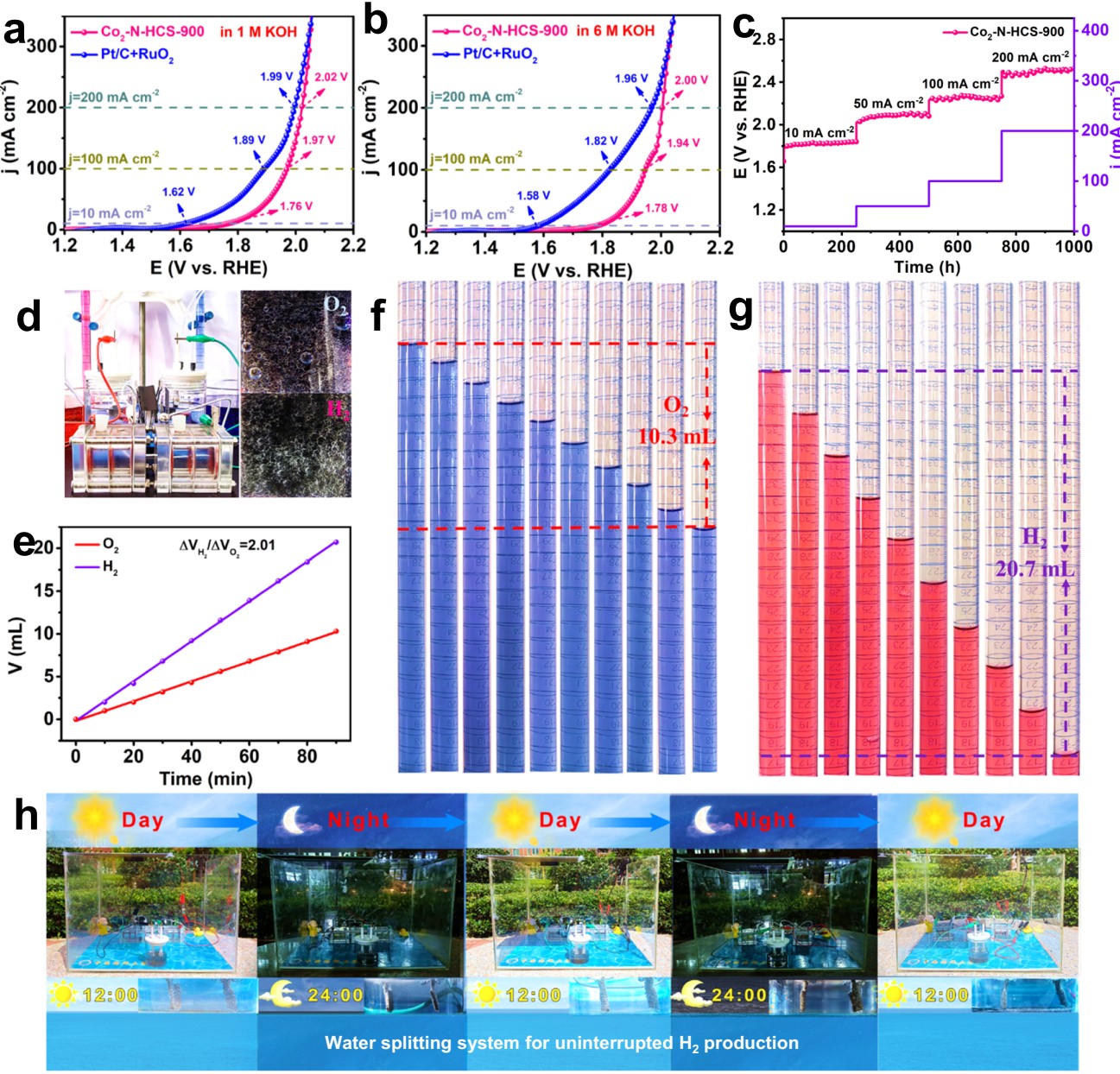

**Fig. 6 | Electrocatalytic overall water-splitting performance. a, b** LSV curves in 1.0 M KOH and 6.0 M KOH. **c** Stability test for overall water splitting cells. **d** Digital image of the water splitting powered by ZABs. **e** Volumes of $O_2$ and $H_2$ evolved with time. **f, g** Enlarged digital images of the measured gas quantities generated at 10 min intervals for 90 min. **h** Photograph of series WSS driven by $Co_2$-N-HCS-900 during day and night over 48 h.

densities of 10, 100, and 200 mA cm$^{-2}$, respectively, which can compete with those of commercial $Pt/C + RuO_2$ counterparts (-1.62, 1.89, and 1.99 V, Fig. 6a). Figure 6b displays the overall water-splitting performance of the $Co_2$-N-HCS-900-based devices in a 6 M KOH solution. A large potential is required to achieve a current density of 10 mA cm$^{-2}$ in comparison with its commercial $Pt/C + RuO_2$ counterparts, but the potential gaps gradually become narrower as the current densities increase from 10 to 200 mA cm$^{-2}$, again emphasizing its good overall water splitting activity. In addition, the $Co_2$-N-HCS-900-based splitting devices maintained a constant potential along with negligible changes after continuous operation for 1000 h in 1 M KOH (Fig. 6c), underlining the excellent stability of $Co_2$-N-HCS-900. Inspired by its excellent trifunctional ORR/OER/HER performance, a cell including two tandem ZABs and water electrolysis devices was established using only $Co_2$-N-HCS-900 (Fig. 6d). The cell could provide an ultra-high $H_2$ production rate of 616 μmol h$^{-1}$, which was determined by the volume of generated

$H_2$ (20.7 mL) and $O_2$ (10.3 mL) over a total of 90 min (Fig. 6e–g). The quantified generated $H_2/O_2$ ratio was calculated to be approximately 2.01, which agrees well with the theoretical ratio (Fig. 6e). As the proof-of-concept, a highly efficient and durable solar-powered WSS that integrates polycrystalline Si solar panels, three tandem ZABs (with an OCP of 4.35 V, Figure S69), and water electrolysis devices was also constructed. The WSS could ensure uninterrupted $H_2$ production for 48 h, throughout day and night (Fig. 6h and S69), showcasing the promising potential of $Co_2$-N-HCS-900 for uninterrupted, large-scale $H_2$ production.

In summary, a NP-to-SA-to-DA atomization and sintering strategy was implemented to achieve a controllable adjustment of the existing configuration states from Co nanoparticles to $CoN_4$ single atoms to $Co_2N_5$ dual atoms at the atomic level. We discovered that dual atom $Co_2N_5$ with low spin magnetic moments disrupts the conventional $\Delta G_{OOH^*}$ to $\Delta G_{OH^*}$ scaling relations, achieving

optimal O* adsorption and moderated H* adsorption/desorption. As expected, Co₂-N-HCS-900 exhibited enhanced multifunctional ORR/OER/HER activity, which enabled the solar-powered WSS to produce $H_2$ continuously over 48 h. More importantly, this universal strategy can be broadened to transform 22 types of conventional s-, p-, and d-block metals or their alloys into dual atom structures. This work both provides a systematic investigation of the formation mechanisms of dual atom catalysts and emphasizes a universal strategy to synthesize dual atom catalysts in pursuit of a breakthrough in multifunctional activities, motivating the rational design of highly efficient multifunctional dual atom catalysts for application in renewable energy conversion technologies.

## Methods

### Chemicals

Hexamethylenetetramine (HMT), 2,4-dihydroxybenzoic acid (DA), sodium oleate (SO), cobalt chloride hexahydrate ($CoCl_2 \cdot 6H_2O$), nickel chloride hexahydrate ($NiCl_2 \cdot 6H_2O$), iron chloride hexahydrate ($FeCl_3 \cdot 6H_2O$), manganese acetate ($Mn(CH_3COO)_2 \cdot 4H_2O$), copper chloride dihydrate ($CuCl_2 \cdot 2H_2O$), zinc acetate dihydrate ($Zn(CH_3COO)_2 \cdot 2H_2O$), antimony trichloride ($SbCl_3$), calcium acetate monohydrate ($Ca(CH_3COO)_2 \cdot H_2O$), bismuth nitrate pentahydrate ($Bi(NO_3)_3 \cdot 5H_2O$), chromium chloride hexahydrate ($CrCl_3 \cdot 6H_2O$), cerium nitrate hexahydrate ($Ce(NO_3)_3 \cdot 6H_2O$), ruthenium(III) chloride anhydrous ($RuCl_3$), aluminum chloride hexahydrate ($AlCl_3 \cdot 6H_2O$), melamine, and n-pentane were obtained from Aladdin Chemical Co. Pluronic P123 (nonionic triblock copolymer, $EO_{20}PO_{70}EO_{20}$) and $RuO_2$ were purchased from Sigma-Aldrich. A commercial Pt/C catalyst (20 wt%) was acquired from Johnson Matthey Co. (Shanghai, China).

### Preparation of hollow polymer spheres (HPS)

The HPS precursor was prepared using the following method[57,58]. An aqueous solution A containing 0.375 mM Pluronic P123 and 12 mM SO was injected into aqueous solution B containing 8.3 mM HMT and 20 mM DA. The mixed solution was then transferred to a Teflon-lined stainless-steel autoclave and heated to 160 °C for 2 h. Finally, the HPS precursors were obtained by centrifuging with water and dried at 50 °C for 24 h.

### Synthesis of Co₂N₅ dual atom on the N-doped hollow carbon spheres (Co₂-N-HCS-900)

HPS (50 mg) was dispersed in n-pentane (10 mL) by ultrasonication for 30 min and stirred for 30 min at room temperature. A certain amount of $CoCl_2 \cdot 6H_2O$ solution was injected into the HPS solution and then continuously stirred for 12 h at room temperature to evaporate the solvent. The precursor (denoted as Co-HPS) was obtained after drying in a vacuum at 50 °C for 24 h. Finally, the Co-HPS precursor and melamine with the ratio of 1:10 were ground together and placed into a tube furnace, heated to 900 °C, and maintained there for 180 min under $N_2$ atmosphere to yield the final catalysts with 20% yield of Co-HPS (termed Co₂-N-HCS-900).

### Synthesis of CoN₄ single atom on the N-doped hollow carbon spheres (Co_SA-N-HCS-900)

Co-HPS precursor and melamine with the ratio of 1:10 were ground together and then placed into a tube furnace, heated to 900 °C, and maintained there for 30 min under $N_2$ atmosphere to yield the Co_SA-N-HCS-900 catalysts (around 20% yield of Co-HPS).

### Synthesis of Co nanoparticles on the hollow carbon spheres (Co_NP/HCS-900)

The Co-HPS precursor was directly placed into a tube furnace, heated to 900 °C, and maintained there for 180 min under $N_2$ atmosphere to yield Co_NP/HCS-900 (around 20% yield of Co-HPS).

### Synthesis of hollow carbon spheres (HCS-900)

The HPS precursor was directly placed into a tube furnace, heated to 900 °C, and maintained there for 180 min under $N_2$ atmosphere to yield HCS-900 (around 20% yield of HPS).

### Preparation of 21 different dual atom catalysts

The Al₂-N-HCS-900, Ca₂-N-HCS-900, Cr₂-N-HCS-900, Mn₂-N-HCS-900, Fe₂-N-HCS-900, Ni₂-N-HCS-900, Cu₂-N-HCS-900, Zn₂-N-HCS-900, Ru₂-N-HCS-900, Sb₂-N-HCS-900, Ce₂-N-HCS-900, Bi₂-N-HCS-900, CoFe-N-HCS-900, CoNi-N-HCS-900, CoCu-N-HCS-900, CoZn-N-HCS-900, CoMn-N-HCS-900, FeNi-N-HCS-900, FeCu-N-HCS-900, FeZn-N-HCS-900, and FeMn-N-HCS-900 were synthesized using a similar procedure to that used for fabricating Co₂-N-HCS-900, except the metal Co salts solution were replaced with a $AlCl_3 \cdot 6H_2O$, $Ca(CH_3COO)_2 \cdot H_2O$, $CrCl_3 \cdot 6H_2O$, $Mn(CH_3COO)_2 \cdot 4H_2O$, $FeCl_3 \cdot 6H_2O$, $NiCl_2 \cdot 6H_2O$, $CuCl_2 \cdot 2H_2O$, $Zn(CH_3COO)_2 \cdot 2H_2O$, $RuCl_3$, $SbCl_3$, $Ce(NO_3)_3 \cdot 6H_2O$, or $Bi(NO_3)_3 \cdot 5H_2O$ solution or a two-mixture solution (1:1).

### Characterization

Transmission electron microscopy (TEM) was performed using a Hitachi H-7650 microscope. High-resolution TEM (HRTEM), high-angle annular dark-field scanning transmission electron microscopy (HAADF-STEM), and energy-dispersive X-ray spectroscopy (EDS) were performed using a JEM-2100F microscope. Aberration-corrected (AC) HAADF-STEM was employed on a Titan-Cubed Themis G2. X-ray absorption near edge structure (XANES) and extended X-ray absorption fine structure (EXAFS) measurements of the Co K-edge were performed in the fluorescence mode at beamline BL14W1. The X-ray diffraction (XRD) patterns were recorded using a Bruker D8 Advance diffractometer. Raman spectra were obtained using a Thermo Fisher spectrometer equipped with helium-neon (633 nm) and argon (532 nm) lasers. X-ray photoelectron spectroscopy (XPS) was conducted using a Thermo ESCALAB 250XI instrument. Ultraviolet photoelectron spectroscopy (UPS) was performed using a Thermo ESCALAB Xi+ instrument equipped with an ultraviolet photoelectron spectroscope (HeI (21.22 eV)). Inductively coupled plasma-optical emission spectrometry (ICP-OES) was used to precisely detect the Co content. Brunauer–Emmett–Teller (BET) analysis was used to investigate the specific surface areas of the catalysts.

### Electrochemical measurements

The ORR, OER, and HER performances were investigated using an electrochemical station (CHI-660E) equipped with a conventional three-electrode system. For the OER and HER, the catalyst-modified pretreated Ni foam, graphite rods, and Hg/HgO electrodes were used as the working, counter, and reference electrodes, respectively. A total of 10 mg catalysts were dispersed into 1 mL Nafion-solution containing water, isopropanol, and Nafion (v/v/v = 4:1:0.1), and then ultrasonicated for 1 h. Then, 100 µL of the suspension was pipetted onto pre-treated Ni foam (1 cm²) and dried under an infrared lamp; this was used as the working electrode with a loading amount of 1 mg cm⁻². Linear sweep voltammetry (LSV) was used to evaluate the OER and HER performances of the catalysts in 1 M KOH with iR compensation. Long-term stability tests were conducted using chronopotentiometric measurements. The overall water-splitting performance of the two-electrode electrolysis devices was investigated in 1.0 and 6.0 M KOH solutions.

For the ORR, a catalyst-modified glassy carbon electrode (GCE), a Pt wire, and a Ag/AgCl (KCl-saturated) electrode were used as the working, counter, and reference electrodes, respectively. 2 mg catalysts were added to 1 mL Nafion-solution and ultrasonicated for 1 h. Then, 27 µL of the suspension was pipetted onto a polished rotating disk electrode (RDE, diameter: 4 mm) or rotating ring disk electrode (RRDE, diameter: 4 mm), which was used as the working electrode with a loading amount of 0.43 mg cm⁻². Linear sweep voltammetry (LSV)

was used to explore the ORR performance in $O_2$-saturated 0.1 M KOH at a rotation rate of 1600 rpm.

## Aqueous Zn-air battery assembly

Homemade aqueous zinc-air batteries (ZABs) were established to assess their practical applications. Polished zinc foil was used as the anode, and a hydrophilic carbon fiber paper substrate coated with a catalyst layer (1 mg cm$^{-2}$) was used as the air cathode. A solution of 6.0 M KOH + 0.2 M $Zn(CH_3COO)_2$ was used as the electrolyte in the ZABs. LSV measurements were performed on a CHI-660 electrochemical workstation at a scan rate of 10 mV s$^{-1}$ at room temperature. The galvanostatic charge and discharge measurements were performed at room temperature by a LAND testing system at 5 mA cm$^{-2}$ with 5 min of discharge followed by 5 min of charge.

## Computational methods

All calculations were performed within the framework of the density functional theory (DFT) as implemented in the Vienna Ab initio simulation package (VASP 5.4.4) code within the Perdew–Burke–Ernzerhof (PBE) generalized gradient approximation and projected augmented wave (PAW) method[34,59–61]. The cut-off energy for the plane-wave basis set was set to 400 eV. An ultrasoft pseudopotential was employed to describe the interaction between the valence electrons and ionic core. The Brillouin zone was sampled using gamma-centered $1 \times 1 \times 1$ k-point meshes to perform geometry optimization and electronic structure calculations. During the geometry optimization, all atoms were allowed to relax without any constraints until the convergence thresholds of maximum force and energy were smaller than 0.05 eV/Å and $1.0 \times 10^{-4}$ eV/atom, respectively. A vacuum layer of 30 Å was introduced to avoid interactions between periodic images. In addition, van der Waals (vdW) interactions were described by Grimme's DFT-D3 scheme with the application of dispersion correction[62]. Furthermore, transition states were searched using the climbing image nudged-elastic-band (CI-NEB) method combined with the VTST code[37,63].

## Data availability

All the data supporting this study are available in the paper and Supplementary Information. Additional data related to this study are available from the corresponding authors on reasonable request. Source data are provided with this paper.

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

## Acknowledgements

M.H. acknowledges the National Natural Science Foundation of China (22279124) and H.J. acknowledges the Natural Science Foundation of Shandong Province (ZR2020ZD10). J.L. and L.X. acknowledges the National Research Foundation of Korea (NRF) grant funded by the Korea government (MSIT) (Grant NRF-2020R1A2C3004146, RS-2023-00235596, RS-2023-00243788). M.G. acknowledges the National Key Research and Development Project (2022YFA1503900, 2022YFA1203400), Guangdong scientific program with contract no. 2019QN01L057. The authors also acknowledge the staff of beamline BL14W1 at the Shanghai Synchrotron Radiation Facility for their support in the XAFS measurements and the Pico Center at SUSTech CRF that receives support from Presidential fund and Development and Reform Commission of Shenzhen Municipality.

## Author contributions

X.W., L.X., C.L. and C.Z. contributed equally. X.W., C.L. and C.Z. performed the experiments, collected the data, analyzed the data, and wrote the manuscript. L.X. conducted the DFT calculations and analyzed the data. H.Y. and R.X. collected the data, analyzed the data. P.C. and X.Z. performed the XAFS measurements. M.G., J.L., H.J. and M.H. designed the experiments, analyzed the data, and wrote the manuscript.

## Competing interests

The authors declare no competing interests.
