## [Peer Review File · Nature Communications]

Establishing libraries for transforming nanoparticles to married dual-atom catalystsREVIEWER COMMENTS

Reviewer #1 (Remarks to the Author):

The manuscript synthesized a series of dual-atom catalysts and disclose the the relationship between spin states and adsorption/desorption of intermediates. The authors reported the controllable regulation from Co nanoparticles to CoN₄ single-atom to Co₂N₅ dual-atom using atomization/sintering strategy via N-stripping/thermal-migrating process. Though the conception sounds interesting, data and analysis are inadequate to test the hypothesis and support the conclusions. The scientific significance of this manuscript can't meet the stringent standard of Nature Communications at current stage. Some additional comments are as follows.

1. For the "Further support for this phenomenon can be demonstrated by the downshift of d band center of Co-3d orbitals from NPto-SA-to-DA, accompanied with the corresponding value of -0.71, -1.61 and -2.27 eV (Fig. 1e)", the figure is inconsistent with description.
2. The authors deem that tailing spin states could optimize adsorption/desorption of intermediates, however, no direct evidences support this opinion. Capacity of samples for calculation is too small to establish correlation between spin states and intermediates adsorption/desorption.
3. Although the XRD pattern of CoNP-HCS-900 prove the existence of metal Co, no evidences could exhibit the morphology of nanoparticals. Please show more charactizations of CoNP-HCS-900.
4. The differences between CoSA-N-HCS-900 and Co₂-N-HCS-900 are not obvious, please show more charactizations and analysis about the distinction of Co single-atom and Co dual-atom.
5. Only XPS of C and N are included in this manuscript, please added comparisons of XPS for all elements of CoNP-HCS-900, CoSA-N-HCS-900 and Co₂-N-HCS-900.
6. The manuscript dedicated to disclose the relationship between spin states and adsorption/desorption of intermediates, but analysis and evidences are insufficient. The authors should exhibit comprehensive descriptions and supply related expriments data.
7. The average coordinated numbers of Co-N/O bond are 4.0 and 3.9 of CoSA-N-HCS-900 and Co₂-N-HCS-900 respectively. That looks pretty much the same. How did the authors conclude CoN₄ structure for CoSA-N-HCS-900 and Co₂N₅-O structure for Co₂-N-HCS-900 from the average coordinated number?

Reviewer #2 (Remarks to the Author):

Comments to the Author

Recommendation: Publish after major revisions noted.

The rational design of efficient dual-atom catalysts with multifunctionality is still in a blind area and full of challenges. In this work, Huang and co-works achieved the controllable regulation from Co nanoparticles to CoN₄ single-atom to Co₂N₅ dual-atom using atomization/sintering strategy via N-stripping/thermal-migrating process, realizing superior multifunctional activities. However, this study has methodological problems and limited innovation, thus the paper is recommended to be major revised. The detailed comments are as follows:

1. Why Co₁₀ is taken as nanoparticles sample for research?
2. In different reaction systems, how to judge the adsorption of catalyst to reaction intermediates should be strengthened or weakened? Considering the combination cannot be too strong or too weak according to Sabatier principle, thus what is the appropriate Ed value?
3. The common conclusion of previously reported researches is that the increase of magnetic moment is conducive to the improvement of catalytic activity. However, the relationship between the size of magnetic moment and the level of activity in this work is inconsistent with it. Please make corresponding comparisons and explain accordingly.
4. How to experimentally verify the process of atomization of nanoparticles to form a single atom of Co? According to the paper, the reaction conditions and raw materials for the synthesis of Co nanoparticles (CoNP-HCS-900) are different from those for the CoSA-N-HCS-900, so this conclusion cannot be strongly verified.
5. Please the author to confirm the relationship between the valence band maximum and the work function in Figure 4f or whether the meaning expressed is the same.
6. This work emphasizes that Co₂N₅ has excellent ORR/OER/HER performance, but the performance test results do not have advantages over the currently reported monoatomic catalysts. Please explain the research significance and innovation of this work compared with other work.
7. In order to realize practical application, the long cycle life of Zn-air battery is generally tested under high current density. Please specify the test current density of this work (Fig. S47).
8. For the convenience of readers' reading and understanding, please improve the clarity of the diagram.
9. Some very related M-N-C works about spin state regulation should be cited, such as Nat. Commun., 2021, 12, 1734, Adv. Funct. Mater., 2022, 2113191, Energy Environ. Sci., 2018, 11, 2348-2352., Adv. Sci., 2021, 2102915., Energy Environ. Sci., 2022, 15, 771-779., ACS Catal., 2021, 11, 12754-12762., Angew. Chem. Int. Ed.2022,61,e2022059.

Reviewer #3 (Remarks to the Author):

In this paper, a NP-to-SA-to-DA atomization and sintering strategy were adopted to achieve the controllable adjustment of the existing configuration states from nanoparticles to single-atom to dual-atom at the atomic level, which was broadened to 22 different elements. Among all the catalysts, the Co dual-atom catalyst (Co₂-N-HCS-900) affords the boosted multifunctional ORR/OER/HER activities, which enables the solar-powered WSS with uninterrupted large-scale H₂ production throughout day and night over 48 h. However, there are still some comments needed to be addressed before it can be published in Nature Communications.

1. The author claimed that the single-atom structure is MN₄, while the dual-atom structure is M₂N₅. However, according to EXAFS fitting results (Table S1) and EDS mapping (Fig. 2), oxygen is dispersed in the sample. The author should try to identify the coordination environment of SACs and DACs.
2. The initial state of SACs in the calculation (Fig.1b), two metal atoms are adjacent. However, in the sample, the metal atoms are randomly dispersed. The author should prove that the metal atoms, which are separated by several carbon atoms, will also become dual-atom structures.

3. The actual metal loading amount should be measured, including nanoparticles, single-atom catalysts, and dual-atom catalysts. Therefore, the mass activity of HER, OER, and ORR should be compared as well.

Reviewer #1 (Remarks to the Author):

The manuscript synthesized a series of dual-atom catalysts and disclose the relationship between spin states and adsorption/desorption intermediates. The authors reported the controllable regulation from Co nanoparticles to CoN₄ single-atom to Co₂N₅ dual-atom using atomization/sintering strategy via N-stripping/thermal-migrating process. Though the conception sounds interesting, data and analysis are inadequate to test the hypothesis and support the conclusions. The scientific significance of this manuscript can't meet the stringent standard of Nature Communications at current stage. Some additional comments are as follows.

Response:

We thank the referee for the positive and critical comments. Keeping the reviewer's suggestions in mind, we have responded point by point to the reviewer's comment as listed in this letter. The changes have been highlighted with yellow in the revised manuscript and supporting information.

Question 1:

For the “Further support for this phenomenon can be demonstrated by the downshift of d band center of Co-3d orbitals from NP-to-SA-to-DA, accompanied with the corresponding value of -0.71, -1.61 and -2.27 eV (Fig. 1e)”, the figure is inconsistent with description.

Response:

We greatly appreciate your insightful comments. We are very sorry for our careless mistake in the original manuscript. According to your valuable suggestions, we have corrected this section.

To make it clear, the related part is listed as follows and highlighted in yellow.

“Further support for this phenomenon can be demonstrated by the downshift of d band center of Co-3d orbitals from NP-to-SA-to-DA, accompanied with the corresponding value of -0.71, -1.61 and -2.27 eV (Figure 1d).”

(Please find the details in Page 6 of our revised manuscript.)

Question 2:

The authors deem that tailing spin states could optimize adsorption/desorption of intermediates, however, no direct evidences support this opinion. Capacity of samples for calculation is too small to establish correlation between spin states and intermediates adsorption/desorption.

Response:

We greatly appreciate your insightful comments. According to your kind suggestions, we have checked the capacity of samples for the calculation to establish correlations. Indeed, as what you have mentioned, the more the capacity of samples, the better the correlation is established. The capacity of three samples for calculation might be small to establish correlation between spin states and intermediates adsorption/desorption. Actually, there are some previous literatures that establish the correlation by using three investigated samples ((1) *Angew. Chem. Int. Ed.*, **2022**, 134, 202207268. (2) *Adv. Mater.*, **2022**, 34, 2204570. (3) *Nano Lett.*, **2023**, 23, 1505-1513. (4) *ACS Nano*, **2022**, 16, 13223-13231. (5) *Appl. Catal. B: Environ.*, **2023**, 321, 122034. (6) *Nano Energy*, **2023**, 105, 108020. (7) *Adv. Mater.*, **2022**, 34, 2110103. (8) *Chem Catal.*, **2023**, 3, 100532). Therefore, the capacity of three samples for calculation might be small, but might be applicable to establish the correlation between spin states and intermediates adsorption/desorption.

Besides, to support the opinion that tailing spin states could optimize adsorption/desorption of intermediates, we have calculated the adsorption free energy of O* intermediates toward ORR/OER and H* intermediates toward HER for the three Co₁₀ nanoparticles, CoN₄ single-atom, and Co₂N₅ dual-atom models. It can be seen in Figure S17 that the Co₂N₅ dual-atom model affords the moderate O* adsorption energy of 1.89 eV among the investigated three models, decoding that it could achieve the optimized O* adsorption/desorption, thus boosting the ORR/OER activities. As for HER, the highest H* adsorption energy of -0.21 eV can be captured among the three investigated models (Figure S18), indicating that it affords moderate adsorption and desorption of H* intermediate, thus endowing excellent HER activities.

To make it clear, the related part is listed as follows and highlighted in yellow.

“These results decode that the Co₂N₅ dual-atom with the decreased spin magnetic moment

can break the universal ΔG_{OOH^*} to ΔG_{OH^*} scaling relation, thus achieving the ideal balanced O^* adsorption. Such phenomenon can be further supported by the moderate O^* adsorption energy (1.89 eV, Figure S17) of Co_2N_5 dual-atom model among three investigated models, decoding that it could achieve the optimized O^* adsorption/desorption, thus boosting the ORR/OER activities.”

“The Co_2N_5 dual-atom possesses the highest ΔG_{H^*} of -0.04 eV (close to the ideal value of 0 eV) among the three investigated models, validating that the low spin state enables the Co_2N_5 dual-atom with moderate adsorption and desorption of H^* intermediate. Besides, the highest H^* adsorption energy of -0.21 eV can be captured among the three investigated models (Figure S18), again validating that it affords moderate adsorption and desorption of H^* intermediate, thus endowing excellent HER activities.”

Figure S17. Calculated adsorption free energies of O^* intermediates on the three Co_{10} nanoparticles, CoN_4 single-atom, and Co_2N_5 dual-atom models, respectively.

Figure S18. Calculated adsorption free energies of H^* intermediates on the three Co_{10} nanoparticles, CoN_4 single-atom, and Co_2N_5 dual-atom models, respectively

(Please find the details in Page 6 and 7 of our revised manuscript and Page 22-23 of our revised supporting information.)

Question 3:

Although the XRD pattern of $\text{Co}_{\text{NP}}\text{-HCS-900}$ prove the existence of metal Co, no evidences could exhibit the morphology of nanoparticles. Please show more characterizations of $\text{Co}_{\text{NP}}\text{-HCS-900}$.

Response:

We greatly appreciate your insightful comments. According to your valuable suggestions, we have conducted more characterizations (including HRTEM and SAED patterns) of $\text{Co}_{\text{NP}}\text{-HCS-900}$. As shown in Figure S25, numerous obvious nanoparticles were uniformly anchored on the hollow carbon spheres. The lattice distance of the nanoparticles was determined to be 0.203 nm that are attributed to the (111) facet of metallic Co, which can be further demonstrated by the selected area electron diffraction (SAED, Figure S25d) and XRD patterns (Figure S24). The high-angle annular dark-field scanning TEM (HAADF-STEM, Figure S25e-i) and corresponding elemental mapping images demonstrate that Co elements were mainly distributed on the nanoparticle area.

To make it clear, the related part is listed as follows and highlighted in yellow.

Figure S25. (a) TEM, (b and c) HRTEM, (d) SAED, and (e-i) HAADF-STEM and C, O, N, Co elemental mapping images for the $\text{Co}_{\text{NP}}\text{-HCS-900}$.

“Further support for this phenomenon can be provided by their corresponding transmission electron microscope (TEM) and high-resolution TEM (HRTEM) images. As shown in Figure S25, numerous nanoparticles, accompanied with the lattice distance of 0.203 nm indexing (111)

facet of metallic Co, were uniformly anchored on the hollow carbon spheres for $\text{Co}_{\text{NP}}\text{-HCS-900}$. And no obvious Co nanoparticles can be found in both $\text{Co}_{\text{SA}}\text{-N-HCS-900}$ (Figure 2b-i) and $\text{Co}_2\text{-N-HCS-900}$ (Figure 2j-q).”

(Please find the details in Page 9 of our revised manuscript and Page 30 of our revised supporting information.)

Question 4:

The differences between $\text{Co}_{\text{SA}}\text{-N-HCS-900}$ and $\text{Co}_2\text{-N-HCS-900}$ are not obvious, please show more characterizations and analysis about the distinction of Co single-atom and Co dual-atom.

Response:

We greatly appreciate your insightful comments. According to your kind suggestions, we have provided more characterizations and analysis about the distinction between $\text{Co}_{\text{SA}}\text{-N-HCS-900}$ (Co single-atom) and $\text{Co}_2\text{-N-HCS-900}$ (Co dual-atom). To begin with, we have conducted the DFT calculations to investigate the possible structures containing the single Co atom structure and married Co_2 structure (model 1 to 10, Figure S34). Moreover, we have added the comparison between the simulated EXAFS and XANES spectra of the possible structures and the experimental spectra. It is found that the simulated spectra based on the single atom CoN_4 structure (model 9) agree well with the experimental EXAFS and XANES results of $\text{Co}_{\text{SA}}\text{-N-HCS-900}$, indicating that this model 9 is the most likely actual structure in $\text{Co}_{\text{SA}}\text{-N-HCS-900}$ (Figure S35, S36 and Table S2). As for $\text{Co}_2\text{-N-HCS-900}$, the simulated spectra based on the model 2 (binuclear Co_2N_5 configurations with oxygen) agree well with the experimental EXAFS and XANES results, indicating that this model is the most likely actual structure in $\text{Co}_2\text{-N-HCS-900}$, again confirming the formation of married $\text{Co}_2\text{N}_5\text{-OH}$ structure in $\text{Co}_2\text{-N-HCS-900}$ (Figure S37, S38 and Table S3). These results demonstrate that there are some differences between Co single-atom of $\text{Co}_{\text{SA}}\text{-N-HCS-900}$ and Co dual-atom of $\text{Co}_2\text{-N-HCS-900}$.

Actually, to make the differences between $\text{Co}_{\text{SA}}\text{-N-HCS-900}$ and $\text{Co}_2\text{-N-HCS-900}$ more obvious, the projected distance between two adjacent Co atoms has been investigated in our original manuscript and supporting information. It can be seen in Figure 2h-i and S29 that the

projected distance between two adjacent Co atoms in Co_{SA}-N-HCS-900 is mainly distributed in the range of 0.30-0.50 nm, indicating that the Co species mainly exist as single atoms. As for Co₂-N-HCS-900 (Figure 2o), a large proportion of Co atoms are adjacent to each other and presented in a form of dual Co atoms, accompanied with the Co-Co distance ranging from 0.12 nm to 0.25 nm (Figure 2p-q and S30), demonstrating the existence of married Co₂ dual-atom dimers. Beyond that, the comparison of the q-space magnitudes in Figure 4b again confirms the existence of Co-N path and Co-Co path in Co₂-N-HCS-900, while only Co-N path exists in Co_{SA}-N-HCS-900.

To make it clear, the related part is listed as follows and highlighted in yellow.

“To further verify the possible structure of Co_{SA}-N-HCS-900, and Co₂-N-HCS-900, DFT calculations were first to investigate the possible structures containing the single Co atom structure and married Co₂ structure (model 1 to 10, Figure S34). Moreover, the comparison between the simulated EXAFS and XANES spectra of the possible structures and the experimental spectra were also recorded. As shown in Figure S35, S36 and Table S2, the simulated spectra based on the single atom CoN₄ model (model 9) agree well with the experimental EXAFS and XANES results of Co_{SA}-N-HCS-900, confirming that this model is the most likely actual structure in Co_{SA}-N-HCS-900. As for Co₂-N-HCS-900, the simulated spectra based on the model 2 (binuclear Co₂N₅ configurations with oxygen) agree well with the experimental EXAFS and XANES results (Figure S37, S38 and Table S3), indicating that this model 2 is the most likely actual structure. These results synergistically validate that CoN₄ single-atom could couple with each other to form married Co₂N₅ dual-atom dimers via thermal migration process.”

Figure S34. The possible Co atomic structures optimized by DFT (green ball: Co; blue ball: N; purple ball: C; red ball: O)

Figure S35. Comparison between the Co K-edge XANES experimental spectrum of Co_{SA}-N-HCS-900 (solid red line) and the theoretical spectrum (solid blue line) calculated with different Co atomic structures (green ball: Co; blue ball: N; purple ball: C; red ball: O).

Figure S36. Comparison between the Co K-edge EXAFS experimental spectrum of Co_{SA}-N-HCS-900 (solid red line) and the theoretical spectrum (solid black line) calculated with different Co atomic structures (green ball: Co; blue ball: N; purple ball: C; red ball: O).

Figure S37. Comparison between the Co K-edge XANES experimental spectrum of Co₂-N-HCS-900 (solid red line) and the theoretical spectrum (solid blue line) calculated with different Co atomic structures (green ball: Co; blue ball: N; purple ball: C; red ball: O).

Figure S38. Comparison between the Co K-edge EXAFS experimental spectrum of Co₂-N-HCS-900 (solid red line) and the theoretical spectrum (solid black line) calculated with different Co atomic structures.

Table S2. The XANES and EXAFS fitting parameters at the Co K-edge of the possible structures from model 1 to model 10 for the Co_{SA}-N-HCS-900.

Model	XANES (S_0^2)	EXAFS (R-factor)
1	10.73	0.1370
2	3.42	0.0151
3	5.34	0.0403
4	12.76	0.1071
5	2.40	0.0737
6	1.51	0.1543
7	7.93	0.0195
8	6.39	0.0192
9	0.46	0.0068
10	4.68	0.0108

Table S3. The XANES and EXAFS fitting parameters at the Co K-edge of the possible structures from model 1 to model 10 for the Co₂-N-HCS-900.

Model	XANES (S_0^2)	EXAFS (R-factor)
1	13.50	0.1603
2	0.39	0.0033
3	6.34	0.0441

4	13.07	0.1184
5	3.31	0.0810
6	2.42	0.1825
7	7.73	0.0267
8	7.30	0.0254
9	3.58	0.0389
10	5.20	0.0128

(Please find the details in Page 12 of our revised manuscript and 39-43 ,51, 72 and 73 of our revised supporting information.)

Question 5:

Only XPS of C and N are included in this manuscript, please added comparisons of XPS for all elements of C_{ONP}-HCS-900, C_{OSA}-N-HCS-900 and C_{O₂}-N-HCS-900.

Response:

We greatly appreciate your insightful comments. According to your valuable suggestions, we have added the comparisons of XPS for all elements (additional O1s and Co 2p XPS spectra) of C_{ONP}-HCS-900, C_{OSA}-N-HCS-900 and C_{O₂}-N-HCS-900 in the revised manuscript. As shown in Co 2p XPS spectra (Figure S45), four peaks attributed Co-N and satellite peaks can be captured for three investigated catalysts, whereas two peaks indexing Co⁰ species can be observed in the C_{ONP}-HCS-900, indicating the existence of metallic Co nanoparticles in C_{ONP}-HCS-900. The O 1s XPS spectra in Figure S46 display those three peaks ascribed to C=O (ca. 531.8 eV), COOH (ca. 533.3 eV) and absorbed water (ca. 536.3 eV) are presented in C_{ONP}-HCS-900, C_{OSA}-N-HCS-900 and C_{O₂}-N-HCS-900. Interestingly, a new peak appears at 530.2 eV that is indexed into the Co-O bond for the C_{O₂}-N-HCS-900, decoding that the Co-O bond was presented in the catalysts.

To make it clear, the related part is listed as follows and highlighted in yellow.

Figure S45. Co 2p high-resolution XPS spectra of (a) Co_{NP} -HCS-900, (b) $\text{Co}_{\text{SA-N}}$ -HCS-900, and (c) Co_2 -N-HCS-900.

Figure S46. O 1s high-resolution XPS spectra of (a) Co_{NP} -HCS-900, (b) $\text{Co}_{\text{SA-N}}$ -HCS-900, and (c) Co_2 -N-HCS-900.

“The Co 2p XPS spectra in Figure S45 shows that four peaks attributed Co-N species and corresponding satellite peaks can be captured for three investigated catalysts, while two peaks indexing Co^0 species can be observed in the Co_{NP} -HCS-900, indicating the existence of metallic Co nanoparticles in Co_{NP} -HCS-900. As displayed in Figure S46, the deconvolution of the O 1s spectra demonstrated the coexistence of oxygen-containing functional groups (C=O at ca. 531.8 eV, COOH at ca. 533.3 eV and absorbed water at ca. 536.3 eV) in Co_{NP} -HCS-900, $\text{Co}_{\text{SA-N}}$ -HCS-900 and Co_2 -N-HCS-900. Interestingly, a new peak appears at 530.2 eV that is indexed into the Co-O bond for the Co_2 -N-HCS-900, decoding that the Co-O bond was presented in the catalysts.”

(Please find the details in Page 14 of our revised manuscript and Page 50-51 of our revised supporting information.)

Question 6:

The manuscript dedicated to disclose the relationship between spin states and adsorption/desorption of intermediates, but analysis and evidences are insufficient. The authors should exhibit comprehensive descriptions and supply related experiments data.

Response:

We greatly appreciate your insightful comments. According to your helpful suggestions, we have provided the comprehensive descriptions and supplied related experiment data to disclose the relationship between spin states and adsorption/desorption of intermediates. According to your Question 2, we have added adsorption/desorption of intermediates by DFT calculations to disclose the relationship between spin states and adsorption/desorption of intermediates. Besides, it is well known that the EPR measurement is a powerful tool to investigate the paramagnetic properties of the catalyst. According to your valuable suggestions, we have conducted the X-band electron paramagnetic resonance (EPR) to investigate the changes of paramagnetic properties from the Co nanoparticles ($\text{Co}_{\text{NP}}\text{-HCS-900}$) to single CoN_4 sites ($\text{Co}_{\text{SA}}\text{-N-HCS-900}$) to married Co_2N_5 sites ($\text{Co}_2\text{-N-HCS-900}$). It can be seen in Figure S47 that a strong signal can be detected for the $\text{Co}_{\text{NP}}\text{-HCS-900}$, indicating that it is paramagnetic. A downward trend can be captured from the $\text{Co}_{\text{NP}}\text{-HCS-900}$ to $\text{Co}_{\text{SA}}\text{-N-HCS-900}$ to $\text{Co}_2\text{-N-HCS-900}$, indicating that the spin magnetic moment was decreased from Co nanoparticles to single CoN_4 sites to married Co_2N_5 sites. Notably, no signal appears for the $\text{Co}_2\text{-N-HCS-900}$, possibly due to the formation of the binuclear Co structure with antiferromagnetic coupling sites ((1) *ACS Catal.*, **2019**, 9, 6588; (2) *J. Coord. Chem.*, **2010**, 56, 467; (3) *J. Coord. Chem.*, **2006**, 59, 3, 255), again confirming that the married Co_2N_5 structure can be successfully constructed in the $\text{Co}_2\text{-N-HCS-900}$. These above-mentioned analysis and evidence could be conducive to disclose the relationship between spin states and adsorption/desorption of intermediates.

To make it clear, the related part is listed as follows and highlighted in yellow.

Figure S47. X-band electron paramagnetic resonance (EPR) spectrum of $\text{Co}_{\text{NP}}\text{-HCS-900}$,

Co_{SA}-N-HCS-900, and Co₂-N-HCS-900.

“As is well known, the X-band electron paramagnetic resonance (EPR) measurement is a powerful tool to investigate the paramagnetic properties of catalysts. It can be seen in Figure S47 that a strong signal can be detected for the Co_{NP}-HCS-900, indicating that it is paramagnetic. A downward trend can be captured from the Co_{NP}-HCS-900 to Co_{SA}-N-HCS-900 to Co₂-N-HCS-900, indicating that the spin magnetic moment was decreased from Co nanoparticles to single CoN₄ sites to married Co₂N₅ sites. Notably, no signal appears for the Co₂-N-HCS-900, possibly due to the formation of the binuclear Co structure with antiferromagnetic coupling sites, again confirming that the married Co₂N₅ structure can be successfully constructed in the Co₂-N-HCS-900.”

(Please find the details in Page 14 of our revised manuscript and Page 52 of our revised supporting information.)

Question 7:

The average coordinated numbers of Co-N/O bond are 4.0 and 3.9 of Co_{SA}-N-HCS-900 and Co₂-N-HCS-900 respectively. That looks pretty much the same. How did the authors conclude CoN₄ structure for Co_{SA}-N-HCS-900 and Co₂N₅-O structure for Co₂-N-HCS-900 from the average coordinated number?

Response:

We greatly appreciate your insightful comments. Indeed, as what you have mentioned, the average coordinated numbers of Co-N/O bond are 4.0 and 3.9 of Co_{SA}-N-HCS-900 and Co₂-N-HCS-900, respectively. It is hard to conclude CoN₄ structure for Co_{SA}-N-HCS-900 and Co₂N₅-O structure for Co₂-N-HCS-900 from the average coordinated number. Thanks again for your insightful comments. In order to identify the coordination environment of Co_{SA}-N-HCS-900 and Co₂-N-HCS-900, we have conducted the DFT calculations to investigate the possible structures containing the single Co atom structure and married Co₂ structure (model 1 to 10, Figure S34), and added the comparison between the simulated EXAFS and XANES spectra of the possible structures and the experimental spectra (Figure S35-S38). It is found that the simulated spectra based on the single atom CoN₄ structure (model 9) agree well with the experimental EXAFS and XANES results of Co_{SA}-N-HCS-900, indicating that this model 9 is the most likely actual structure in Co_{SA}-N-HCS-900 (Figure S35, S36 and Table S2). As

for Co₂-N-HCS-900, the simulated spectra based on the model 2 (binuclear Co₂N₅ configurations with oxygen) agree well with the experimental EXAFS and XANES results, indicating that this model is the most likely actual structure in Co₂-N-HCS-900, again confirming the formation of married Co₂N₅-OH structure in Co₂-N-HCS-900 (Figure S37, S38 and Table S3). These results demonstrate that there are differences between Co single-atom of Co_{SA}-N-HCS-900 and Co dual-atom of Co₂-N-HCS-900.

To make it clear, please find the related revised part in response to Question 4 of this letter.

We really appreciate your valuable and constructive suggestions, and we hope you are satisfied with the present manuscript which is obviously improved with the help from you.

Reviewer #2 (Remarks to the Author):

Comments to the Author

Recommendation: Publish after major revisions noted.

The rational design of efficient dual-atom catalysts with multifunctionality is still in a blind area and full of challenges. In this work, Huang and co-workers achieved the controllable regulation from Co nanoparticles to CoN₄ single-atom to Co₂N₅ dual-atom using atomization/sintering strategy via N-stripping/thermal-migrating process, realizing superior multifunctional activities. However, this study has methodological problems and limited innovation, thus the paper is recommended to be major revised. The detailed comments are as follows:

Response:

We thank the referee for the positive and critical comments. Keeping the reviewer's suggestions in mind, we have responded point by point to the reviewer's comment as listed in this letter. The changes have been highlighted with yellow in the revised manuscript and supporting information.

Question 1:

Why Co₁₀ is taken as nanoparticles sample for research?

Response:

We greatly appreciate your insightful comments. Actually, based on the previously reported literature, the M₁₀ model (M=Pd, Pt, Ni) was taken as the nanoparticles sample for the research ((1) *Nat. Nanotech.*, **2018**, 13, 856-861. (2) *Nat. Nanotech.*, **2019**, 14, 851-857. (3) *J. Am. Chem. Soc.*, **2021**, 143, 18643-18651. (4) *Angew. Chem. Int. Ed.*, **2023**, 202218630). Therefore, according to the above-mentioned literature, the Co₁₀ is taken as the nanoparticle sample for research in our manuscript.

To make it clear, the related part is listed as follows and highlighted in yellow.

“Density functional theory (DFT) was conducted to reveal the structural transformation mechanism from nanoparticles to single-atom and then to dual-atom (NP-to-SA-to-DA) via taking the Co₁₀ nanoparticles as examples. ^{36, 37}”

36. Wei SJ, *et al.* Direct observation of noble metal nanoparticles transforming to thermally

stable single atoms. *Nat. Nanotechnol.* **13**, 856-861 (2018).

37. Chen YJ, *et al.* Thermal atomization of platinum nanoparticles into single atoms: An effective strategy for engineering high-performance nanozymes. *J. Am. Chem. Soc.* **143**, 18643-18651 (2021).

(Please find the details in Page 4 and 23 of our revised manuscript.)

Question 2:

In different reaction systems, how to judge the adsorption of catalyst to reaction intermediates should be strengthened or weakened? Considering the combination cannot be too strong or too weak according to Sabatier principle, thus what is the appropriate E_d value?

Response:

We greatly appreciate your insightful comments. In general, the adsorption strength of the catalyst to the reaction intermediates can affect the reaction kinetics and selectivity ((1) *Nat. Energy*, **2020**, 5, 891-899. (2) *Adv. Mater.*, **2022**, 34, 2110604. (3) *Nano Energy*, **2020**, 78, 105128). However, the reaction intermediates are different in different reaction systems, which makes it cannot judge the adsorption of catalyst to reaction intermediates should be strengthened or weakened in different reaction systems ((1) *Angew. Chem. Int. Ed.*, **2022**, 61, 202113664. (2) *Adv. Mater.*, **2022**, 34, 2107421. (3) *Angew. Chem. Int. Ed.*, **2021**, 60, 19262-19271). Therefore, to judge whether the adsorption of catalyst to reaction intermediates should be strengthened or weakened, it is necessary to consider the specific reaction system and the properties of the catalyst and intermediates involved.

Indeed, as what you have mentioned, according to Sabatier principle, the adsorption strength cannot be too strong or too weak ((1) *ACS Catal.*, **2020**, 10, 9086-9097. (2) *Angew. Chem. Int. Ed.*, **2018**, 57, 5076-5080. (3) *Nano Energy*, **2023**, 105, 108020). If the adsorption is too weak, the intermediates may not be held tightly enough to allow for efficient reaction pathways. On the other hand, if the adsorption is too strong, the intermediates may be held too tightly, making it difficult for them to react with other species or desorb from the catalyst surface. **A moderate strength of adsorption is preferred, which can be frequently achieved by adjusting the E_d value to be neither too high nor too low.** However, this E_d value can

vary depending on the specific reaction and the properties of the catalyst. Therefore, the appropriate E_d value is not a fixed value and also depended on the specific reaction system and the properties of the catalyst and intermediates involved.

Question 3:

The common conclusion of previously reported researches is that the increase of magnetic moment is conducive to the improvement of catalytic activity. However, the relationship between the size of magnetic moment and the level of activity in this work is inconsistent with it. Please make corresponding comparisons and explain accordingly.

Response:

We greatly appreciate your insightful comments. Indeed, as what you have mentioned, previous studies have shown that an increase in the magnetic moment could be conducive to the improvement of catalytic activity ((1) *Nat. Commun.*, **2021**, 12, 1734. (2) *Nano Energy*, **2023**, 105, 108020. (3) *Adv. Sci.*, **2021**, 8, 2102915. (4) *Chem*, **2023**, 9, 181-197. (5) *Adv. Mater.*, **2022**, 34, 2202240. (6) *Energy Storage Mater.*, **2022**, 50, 12-20. (7) *Angew. Chem. Int. Ed.*, **2022**, 134, 202117617. (8) *Adv. Energy Mater.*, **2022**, 12, 2103588. (9) *ACS Catal.*, **2021**, 11, 8837-8846.). However, there are some previous researches concluded that the decrease in the magnetic moment could also be conducive to improve catalytic activity ((1) *Energy Environ Sci.*, **2016**, 9, 2418-2432. (2) *Adv. Funct. Mater.*, **2019**, 29, 1906174. (3) *Appl. Catal. B: Environ.*, **2023**, 323, 122163. (4) *Angew. Chem. Int. Ed.*, **2022**, 61, 202114293. (5) *Angew. Chem. Int. Ed.*, **2022**, 61, 202201007. (6) *Angew. Chem. Int. Ed.*, **2021**, 133, 25608-25614. (7) *Appl. Catal. B: Environ.*, **2021**, 285, 119778). Therefore, the decrease of magnetic moment might be also conducive to the improvement of catalytic activity.

To make it clear, we have added the meaning expressed of the work function and valance band maximum in the revised manuscript.

“Based on the DFT analysis, the Co_2N_5 dual-atom owns the tailored spin state that could trigger the moderated and balanced adsorption and desorption of reaction intermediates toward ORR/OER/HER, thus advancing trifunctional performance.^{42, 43}”

42. Lv QL, Zhu Z, Ni YX, Geng JR, Li FJ. Spin-state manipulation of two-dimensional metal-organic framework with enhanced metal-oxygen covalency for lithium-oxygen

batteries. *Angew. Chem. Int. Ed.* **61**, 202114293 (2022).

43. He T, *et al.* Theory-guided regulation of FeN₄ spin state by neighboring Cu atoms for enhanced oxygen reduction electrocatalysis in flexible metal-air batteries. *Angew. Chem. Int. Ed.* **61**, 202201007 (2022).

(Please find the details in Page 7 and 23 of our revised manuscript.)

Question 4:

How to experimentally verify the process of atomization of nanoparticles to form a single atom of Co? According to the paper, the reaction conditions and raw materials for the synthesis of Co nanoparticles (Co_{NP}-HCS-900) are different from those for the Co_{SA}-N-HCS-900, so this conclusion cannot be strongly verified.

Response:

We greatly appreciate your insightful comments. According to your kind suggestions, to experimentally verify the conclusions (atomization of nanoparticles to form a single atom of Co) of our manuscript, we have conducted the additional experiments for preparing the catalysts (named as Co_{NP}-N-HCS-300) by the same raw materials but at the pyrolysis temperature of 300 °C. Moreover, the aberration-corrected high-angle annular dark-field scanning transmission electron microscopy (AC HAADF-STEM) was further conducted to investigate the atomic states of Co species for Co_{NP}-N-HCS-300. As shown in Figure S28, some aggregated nanoparticles (marked with red cycles) can be captured on the carbon substrate, indicating that some Co nanoparticles are surfaced on Co_{NP}-N-HCS-300. As for Co_{SA}-N-HCS-900, a large proportion of Co atoms with the Co-Co distance ranging from 0.30-0.50 nm, indicating that the Co species mainly exist as single atoms. These results can be strongly verified the process of atomization of nanoparticles to form a single Co atom in our manuscript. The possible reason for this phenomenon is that the melamine can be decompose at high temperature (>400 °C) and serve as the N agent that is generally coordinated with Co atoms to form Co-N_x moieties for promoting the Co nanoparticles atomization (*Nat. Commun.*, **2019**, 10, 1278).

To make it clear, the related part is listed as follows and highlighted in yellow.

“Then Co_{NP}-HCS-900 was obtained by directly pyrolyzing the Co-HPS, and the Co_{NP}-N-

HCS-300 are synthesized by pyrolyzing the Co-HPS with melamine at 300 °C. Besides, Co_{SA}-N-HCS-900 can be synthesized when the melamine was added in the above pyrolysis process since the melamine can be decomposed at high temperature (>400 °C) and serve as the N agent that is generally coordinated with Co atoms to form Co-N_x moieties for promoting the Co nanoparticles atomization.⁴⁴”

“Aberration-corrected high-angle annular dark-field scanning transmission electron microscopy (AC HAADF-STEM) was further conducted to investigate the atomic states of Co species for Co_{NP}-N-HCS-300, Co_{SA}-N-HCS-900 and Co₂-N-HCS-900. As shown in Figure S28, some aggregated nanoparticles (marked with red cycles) can be captured on the carbon substrate, indicating that some Co nanoparticles are surfaced on Co_{NP}-N-HCS-300.”

44. Zhao L, *et al.* Cascade anchoring strategy for general mass production of high-loading single-atomic metal-nitrogen catalysts. *Nat. Commun.* **10**, 1278 (2019).

Figure S28. AC HAADF-STEM images of Co_{NP}-N-HCS-300.

(Please find the details in Page 8 and 23 of our revised manuscript and Page 33 of our revised supporting information.)

Question 5:

Please the author to confirm the relationship between the valance band maximum and the work function in Figure 4f or whether the meaning expressed is the same.

Response:

We greatly appreciate your insightful comments. According to your valuable suggestions, we have confirmed the relationship between the valance band maximum (VBM) and the work function (Φ) in Figure 4f. According to the literatures ((1) *J. Mater. Chem. A*, **2019**, 7, 19008-

19016; (2) *ACS Energy Lett.*, **2019**, 4, 534-541; (3) *ACS Appl. Nano Mater.*, **2018**, 1, 3673-3681; (4) *ACS Catal.*, **2020**, 10, 7734-7746; (5) *Angew. Chem. Int. Ed.*, **2020**, 59, 6929-6935; (6) *Chem*, **2023**, 9, 181-197), the work function is determined by the equation $E_{\Phi} = h\nu - (E_{\text{cutoff}} - E_f)$; and the valance band maximum is determined by the equation: $E_{\text{VBM}} = h\nu - (E_{\text{cutoff}} - E_f) + E_{\text{VB}}$; where $h\nu$ (21.22 eV) represents the incident photoenergy from He I excitation source, E_{cutoff} is the secondary electron cutoff level, and E_f is the Fermi energy level.

Besides, the meaning expressed of the work function and valance band maximum is not the same. The work function could be one of the critical factors determining the electron transfer process since it represents the minimum energy for inner electrons to escape from the surface of a catalyst. A lower work function indicates a lower activation barrier for electron donation from the surface of the electrocatalyst to the reactant. A high work function signifies a high activation barrier for the electrons to be transferred to the reactant ((1) *ACS Appl. Nano Mater.*, **2018**, 1, 3673-3681; (2) *Chem*, **2023**, 9, 181-197; (3) *Nat. Commun.*, **2022**, 12, 3036; (4) *Nat. Catal.*, **2019**, 2, 688-695). While the valance band maximum (VBM) is usually referred to as the highest occupied molecular orbital (HOMO), which is related to the highest energy level of the valence band in a solid material. It is an important parameter in determining the electronic properties of catalysts, such as its electrical conductivity. As is well known, the valence electrons near the Fermi level make dominant contributions to the d states, so the valence band shifts represent the movement of the E_d energy level. A higher valance band maximum (VBM) indicates that the valance band gets away from the Fermi level. A lower VBM suggests that the valance band gets close to the Fermi level. ((1) *Chem*, **2023**, 9, 181-197; (2) *ACS Energy Lett.*, **2019**, 4, 534-541; (3) *J. Mater. Chem. A*, **2019**, 7, 19008-19016).

To make it clear, we have added the meaning expressed of the work function and valance band maximum in the revised manuscript.

“Ultraviolet photoemission spectroscopy (UPS) was conducted to investigate the electronic state of the $\text{Co}_{\text{NP}}\text{-HCS-900}$, $\text{Co}_{\text{SA}}\text{-N-HCS-900}$ and $\text{Co}_2\text{-N-HCS-900}$. The work function represents the minimum energy for inner electrons to escape from the surface of catalysts. As shown in Figure 4f and S48, the work function of $\text{Co}_2\text{-N-HCS-900}$ is determined to be 5.04 eV, higher than that of the $\text{Co}_{\text{SA}}\text{-N-HCS-900}$ (4.90 eV) and $\text{Co}_{\text{NP}}\text{-HCS-900}$ (4.52 eV). Besides, the valence band maximum is referred to as the highest occupied molecular

orbital (HOMO), which is related to the highest energy level of the valence band in a solid material. It is well known that the shift of the valence band is indicative of the change of the E_d energy level, because the valence electrons adjacent to the Fermi level play a major contribution to the d states.⁵³ The Co₂-N-HCS-900 affords a high calculated valence band maximum (VBM) value of 5.63 eV compared to Co_{SA}-N-HCS-900 of 5.00 eV, and Co_{NP}-HCS-900 of 4.41 eV (Figure 4f, 4g, and S48), indicating that the valence band gets away from the Fermi level for the Co₂-N-HCS-900. The larger work function and VBM enable the Co₂-N-HCS-900 to pay a higher energetic barrier for donating electrons and a lower E_d energy level, thus resulting in a favorable interaction between the intermediates and active site and enhanced reaction activity.⁵³

(Please find the details in Page 14-15 of our revised manuscript.)

Question 6:

This work emphasizes that Co₂N₅ has excellent ORR/OER/HER performance, but the performance test results do not have advantages over the currently reported monoatomic catalysts. Please explain the research significance and innovation of this work compared with other work.

Response:

We greatly appreciate your insightful comments. According to your kind suggestions, we have added some currently reported monoatomic catalysts for comparison with the single ORR/OER/HER performance. As shown in Table S4, it can be captured the Co₂-N-HCS-900 with the Co₂N₅ structure affords a comparable ORR performance with half-wave potential of 0.86 V with the other reported monoatomic catalysts. Besides, Co₂-N-HCS-900 also pays the lowest OER overpotentials of 333 mV at 10 mA cm⁻² and smallest Tafel slope of 97.1 mV dec⁻¹ in four investigated catalysts, which can approach the commercial RuO₂ and other advanced monoatomic M-N-C catalysts (Figure 5b, 5e and Table S5). As for HER, Co₂-N-HCS-900 achieves good activities in view of its low overpotential (166 mV at 10 mA cm⁻² and 252 mV at 100 mA cm⁻²) and small Tafel slopes (83.9 mV dec⁻¹), which are also comparable to those of most reported monoatomic M-N-C catalysts (Figure 5c, 5f and Table S6). More importantly, we have added the comparison of the multifunctional ORR/OER/HER performance between

the currently reported trifunctional monoatomic catalysts and Co₂-N-HCS-900 (Table S7). It can be found that the Co₂-N-HCS-900 possess the excellent tri-functional ORR/OER/HER performance, demonstrating that the Co₂-N-HCS-900 do have advantages over the currently reported monoatomic catalysts in multifunctionality.

Besides, according to your kind suggestions, we have explained the research significance and innovation of this work compared with other work. The research significance of this manuscript is explained as below:

1. Dual-atom catalyst (DAC), bridging single-atom and metal/alloy nanoparticle catalysts, offers more chances to conquer the challenges and limitations faced by single-atom catalysts via synergistically adjusting the adsorption/desorption behaviors and activation of intermediates, thus accomplishing the accelerated reaction kinetics and efficient multifunctional performance. **Unfortunately, the rational design of the high-efficient and robust DAC with multifunctionality is still in a blind area and full of challenges due to the lack of advanced fundamental knowledge of formation mechanisms for DAC. Herein, we developed an atomization/sintering strategy, i.e., nanoparticle to single-atom to dual-atom (NP-to-SA-to-DA), to realize the controllable adjustment of the existing configuration states at the atomic level. Most strikingly, this NP-to-SA-to-DA atomization and sintering strategy could be broadened to prepare 22 kinds of s-, p-, and d-block metal dual-atom catalyst libraries,** undoubtedly leading to an upsurge in the rational design of efficient and stable dual-atom catalysts for application in energy conversion technologies.

2. This work mainly take Co species as a representative example, and found that the Co₂ dual-atom catalyst with the tailored spin state could achieve the ideal balanced adsorption/desorption of O* and H* intermediates, endowing its efficient multifunctional performance. As expected, the Co₂-N-HCS-900 affords the superior multifunctional ORR/OER/HER activities for comparison with other currently reported multifunctional catalysts. It also enables the Zn-air batteries (ZABs) with splendid cycling charge-discharge stability over 800 h and the water splitting to operate over 1000 h. Moreover, it enables the solar-powered water-splitting system for sustainable hydrogen production throughout the day and night, displaying the promising potential for achieving high-throughput H₂ production.

3. Although the performance test results of Co₂ dual-atom catalyst might not have

advantages over the currently reported monoatomic catalysts, we can use the discovery of this work to guide the design of other dual-atom catalysts with better performance, thus achieving the further optimization of catalyst performance in the future.

To sum up, the present work not only depicts a facile strategy to build up the DAC libraries, but also advance the in-depth scientific understanding of formation mechanisms for DAC. It will be of interest to many fields for the general scientific community.

To make it clear, the related part is listed as follows and highlighted in yellow.

“These above-mentioned results highlight the significant role of adjacent Co atoms in Co₂N₅ structure in realizing the advanced trifunctional activities, which also do have advantages over the currently reported monoatomic catalysts in multifunctionality (Table S7).”

Table S4. Comparison of alkaline ORR performances between Co₂-N-HCS-900 and other M-N-C materials in the literatures.

Catalysts	E _{1/2} /VRHE	Tafel/mV dec ⁻¹	Electrolyte	Reference
Co ₂ -N-HCS-900	0.86	48	0.1 M KOH	This work
Sb1/NG(O)	0.86	54	0.1 M KOH	Angew. Chem. Int. Ed. 61 , 202202200 (2022)
PSTA-Co-1000	0.878		0.1 M KOH	Angew. Chem. Int. Ed. 132 , 14747-14754 (2020)
Fe/Co-N _x -C	0.86	53.6	0.1 M KOH	Small , 16 , 2000742 (2020)
Zn-B/N-C	0.886	50	0.1 M KOH	Angew. Chem. Int. Ed. 133 , 183-187 (2021)
Zn-N-C-1	0.873	103	0.1 M KOH	Angew. Chem. Int. Ed. 58 , 7035-7039 (2019)
Cu/Zn@NC	0.83	54.8	0.1 M KOH	Angew. Chem. Int. Ed. 60 , 14005-14012 (2021)

Co-Te DASs/N-C	0.852	62.3	0.1 M KOH	Small 18 , 2201974 (2022)
FeCoN _x /C	0.86	70	0.1 M KOH	J. Am. Chem. Soc. 141 , 17763-17770 (2019)
PtNPC-0.5	0.87		0.1 M KOH	Angew. Chem. Int. Ed. 60 , 21911-21917 (2021)
Cu-N-C	0.83	37	0.1 M KOH	J. Am. Chem. Soc. 143 , 14530-14539 (2021)
MS-CoSA-N- C-800°C	0.86	88.7	0.1 M KOH	ACS Nano 16 , 11944-11956 (2022)
(Zn, Cu)-NC	0.88	87.1	0.1 M KOH	Adv. Funct. Mater. 32 , 2203471 (2022)
CoN ₄ /NG	0.88	59	0.1 M KOH	Nano Energy 50 , 691-698 (2018)
Co-N- C@F127	0.84		0.1 M KOH	Energy Environ. Sci. 12 , 250-260 (2019)
NiSAs- Pd@NC(2: 1)	0.84	55	0.1 M KOH	J. Mater. Chem. A 10 , 6086-6095 (2022)

Table S5. Comparison of alkaline OER performances between Co₂-N-HCS-900 and other M-N-C materials in the literatures.

Catalysts	η_{10} / mV	Tafel/mV dec ⁻¹	Electrolyte	Reference
Co ₂ -N-HCS-900	333	97.1	1 M KOH	This work
CoFe-N-C	360	67.7	1 M KOH	Nano Lett. 22 , 3392-3399 (2022)
IrFe-N-C	350	43	1 M KOH	ACS Catal. 12 , 9397-9409 (2022)
NCAG/Fe-Cu	380		1 M KOH	Angew. Chem. Int. Ed.

					61 , 2201007 (2022)
FeCo SAs@					ACS Nano
Co/N-GC	290	56.6	1 M KOH		15 , 14683-14696 (2021)
FeCo-DACs/NC	370	82.7	1 M KOH		Adv. Mater. 34 , 2107421 (2022)
Fe ₁ Co ₃ -NC-1100	349	99.93	1 M KOH		ACS Catal. 12 , 1216-1227 (2022)
Fe,Co,N-C	410	76	1 M KOH		ACS Nano 16 , 7890-7903 (2022)
CoSA/NCs	303	76	1 M KOH		Appl. Catal. B- Environ. 316 , 121674 (2022)
CoSA/N-HCS	306	38.1	1 M KOH		Adv. Energy Mater. 10 , 2002896 (2020)
NiFe-DASC	310	45	1 M KOH		Nat. Commun. 12 , 1-11 (2021)
Fe-N/S-CNT-GR	370		0.1 M KOH		ACS Catal. 12 , 7994-8006 (2022)
CoN ₄ /NG	380	81	0.1 M KOH		Nano Energy 50 , 691-698 (2018)
NiSAs- Pd@NC(2: 1)	380	79	0.1 M KOH		J. Mater. Chem. A , 10 , 6086-6095 (2022)
MoS ₂ @Fe-N-C	360	98	0.1 M KOH		P. Natl. A. Sci. 118 , 2110036118 (2021)

Table S6. Comparison of alkaline HER performances between Co₂-N-HCS-900 and other M-N-C materials in the literatures.

Catalysts	η_{10} / mV	Tafel/mV dec ⁻¹	Electrolyte	Reference
-----------	------------------	-------------------------------	-------------	-----------

CO ₂ -N-HCS-900	166	83.9	1 M KOH	This work
Co _{SA} /N, S-HCS	165	96	1 M KOH	Adv. Energy Mater. 10 , 2002896 (2020)
Ru/Co-N-C- 800 °C	19	27.8	1 M KOH	Adv. Mater. 34 , 2110103 (2022)
Ru-1.0	13	25.3	1 M KOH	Adv. Energy Mater. 11 , 2101242 (2021)
S-Co/N/C	121	47	1 M KOH	ACS Catal. 11 , 4498-4509 (2021)
RuSACoFe ₂ /G	164	116	1 M KOH	Energy Environ. Sci. 13 , 5152-5164 (2020)
CuPor-RuN ₃	114		1 M KOH	Adv. Funct. Mater. 31 , 2107290 (2021)
NiCo-SAD-NC	61	55	1 M KOH	Nat. Commun. 12 , 6766 (2021)
Ni-N-C	30.8	32	1 M KOH	J. Am. Chem. Soc. 138 , 14546 (2016)
Co-NMGO	146	95	1 M KOH	Adv. Energy Mater. 11 , 2101619 (2021)
Ru@Co/N-CNTs- 2	48	33	1 M KOH	ACS Sustain. Chem. Eng. 8 , 9136 (2020)
CoFeN- NCNTs//CCM	151	130	1 M KOH	Adv. Funct. Mater. 32 , 2107608 (2021)
Co-Te DASs/N-C	217	76.8	1 M KOH	Small , 18 , 2201974 (2022)
Co/CNFs (1000)	190	56	1 M KOH	Adv. Mater. 31 , 1808043 (2019)

CoSAs-MoS ₂ /TiN NRs	187	53.5	1 M KOH	Adv. Funct. Mater. 31 , 2100233 (2021)
Co-BM-C	126	63	1 M KOH	Chem. Eng. J. 433 , 134089 (2022)

Table S7. Comparison of trifunctional electrocatalytic performance between Co₂-N-HCS-900 and other M-N-C materials in the literatures.

Catalysts	E _{1/2} of ORR	η ₁₀ / mV of OER	η ₁₀ / mV of HER	Reference
Co ₂ -N-HCS-900	0.86	333	166	This work
Co/CNFs (1000)	0.89	320	190	Adv. Mater. 31 , 1808043 (2019)
TSA _s	0.88	393	94	Adv. Energy Mater. 13 , 2203150 (2023)
Co _{SA} /N, S-HCS	0.85	306	165	Adv. Energy Mater. 10 , 2002896 (2020)
Co-N, P-HCS	0.89	320	164	Adv. Mater. 34 , 2204021 (2022)
CF-NG-Co	0.88	400	212	J. Mater. Chem. A 6 , 489-497 (2018)
Co-NC@CC	0.81	240	73	Adv. Funct. Mater. 31 , 2009853 (2021)
Mo-N/C@MoS ₂	0.81	390	117	Adv. Funct. Mater. 27 , 1702300 (2017)
Fe ₃ C-Co/NC	0.88	340	238	Adv. Funct. Mater. 29 , 1901949 (2019)
Co ₂ P/CoNPC	0.84	326	130	Adv. Mater. 32 , 2003649 (2020)

(Please find the details in Page 17 of our revised manuscript and Page 74-77 of our revised

supporting information.)

Question 7:

In order to realize practical application, the long cycle life of Zn-air battery is generally tested under high current density. Please specify the test current density of this work (Fig. S47).

Response:

We greatly appreciate your insightful comments. Actually, the galvanostatic charge and discharge were performed at room temperature by a LAND testing system at 5 mA cm⁻² with 5 min of discharge followed by 5 min of charge. The same procedure can be also adopted by the other reported literature for evaluating the ZABs stability ((1) *Adv. Mater.*, **2021**, 33, 2007525. (2) *Adv. Mater.*, **2019**, 31, 1901666. (3) *Energy Environ. Sci.*, **2022**, 15, 5039-5058. (4) *Appl. Catal. B: Environ.*, **2022**, 317, 121758). According to your valuable suggestions, we have added the test current density of the Zn-air battery in the Fig. S47 of our original supporting information (Figure S61 of our revised supporting information).

In addition, according to your suggestions, we have tested the cycle stability of the Zn-air battery under high current density (50 mA cm⁻²). It can be seen in Figure S63 that the ZABs powered by Co₂-N-HCS-900 can operate over 100 cycles under high current density of 50 mA cm⁻², powerfully demonstrating its practical application.

To make it clear, the related part is listed as follows and highlighted in yellow.

“The ZABs powered by Co₂-N-HCS-900 also possess an excellent round-trip efficiency of 58.1 % **at current density of 5 mA cm⁻²** and distinguished ultralong lifespan over 800 h accompanied with the negligible round-trip efficiency fading (Figure S61 and S62). **Also, the Co₂-N-HCS-900-based ZABs can operate over 600 cycles under high current density of 50 mA cm⁻²** (Figure S63), powerfully demonstrating its promising practical application.”

Figure S61. Galvanostatic discharge-charge cycling curves for ZABs driven by Co₂-N-HCS-900 at current density of 5 mA cm⁻².

Figure S63. Galvanostatic discharge-charge cycling curves for ZABs driven by Co₂-N-HCS-900 at current density of 50 mA cm⁻².

(Please find the details in Page 17 of our revised manuscript and Page 66 and 68 of our revised supporting information.)

Question 8:

For the convenience of readers' reading and understanding, please improve the clarity of the diagram.

Response:

We greatly appreciate your insightful comments. According to your valuable suggestions, we have improved the clarity of Figure 1 - Figure 6 (600 dpi of the pictures).

(Please find the details in Page 5, 8, 10, 13, 16 and 18 of our revised manuscript.)

Question 9:

Some very related M-N-C works about spin state regulation should be cited, such as *Nat. Commun.*, 2021, 12, 1734, *Adv. Funct. Mater.*, 2022, 2113191, *Energy Environ. Sci.*,

2018, *11*, 2348-2352., *Adv. Sci.*, 2021, 2102915., *Energy Environ. Sci.*, 2022, *15*, 771-779., *ACS Catal.*, 2021, *11*, 12754-12762., *Angew. Chem. Int. Ed.*, 2022, *61*, 2022059.

Response:

We greatly appreciate your insightful comments. According to your valuable suggestions, we have cited some very related M-N-C works about spin state regulation in our revised manuscript.

To make it clear, the related part is listed as follows and highlighted in yellow.

13. Xue DP, *et al.* Boron-tethering and regulative electronic states around iridium species for hydrogen evolution. *Adv. Funct. Mater.* **32**, 2113191 (2022).
14. Xia HC, *et al.* Evolution of a solid electrolyte interphase enabled by FeN_x/C catalysts for sodium-ion storage. *Energy Environ. Sci.* **15**, 771-779 (2022).
18. Yin HB, *et al.* Phosphorus-driven electron delocalization on edge-type FeN₄ active sites for oxygen reduction in acid medium. *ACS Catal.* **11**, 12754-12762 (2021).
24. Han YH, *et al.* Electronic structure engineering to boost oxygen reduction activity by controlling the coordination of the central metal. *Energy Environ. Sci.* **11**, 2348-2352 (2018).
32. Yang GG, *et al.* Regulating Fe-spin state by atomically dispersed Mn-N in Fe-N-C catalysts with high oxygen reduction activity. *Nat. Commun.* **12**, 1734 (2021).
33. Zheng XB, *et al.* Ru-Co pair sites catalyst boosts the energetics for the oxygen evolution reaction. *Angew. Chem. Int. Ed.* **61**, 2205946 (2022).
47. Wang YJ, *et al.* Boosting nitrogen reduction to ammonia on FeN₄ sites by atomic spin regulation. *Adv. Sci.* **8**, 2102915 (2021).

(Please find the details in Page 21-24 of our revised manuscript.)

We really appreciate your valuable and constructive suggestions, and we hope you are satisfied with the present manuscript which is obviously improved with the help from you.

Reviewer #3 (Remarks to the Author):

In this paper, a NP-to-SA-to-DA atomization and sintering strategy were adopted to achieve the controllable adjustment of the existing configuration states from nanoparticles to single-atom to dual-atom at the atomic level, which was broadened to 22 different elements. Among all the catalysts, the Co dual-atom catalyst (Co₂-N-HCS-900) affords the boosted multifunctional ORR/OER/HER activities, which enables the solar-powered WSS with uninterrupted large-scale H₂ production throughout day and night over 48 h. However, there are still some comments needed to be addressed before it can be published in Nature Communications.

Response:

We thank the referee for the positive and critical comments. Keeping the reviewer's suggestions in mind, we have responded point by point to the reviewer's comment as listed in this letter. The changes have been highlighted with yellow in the revised manuscript and supporting information.

Question 1:

The author claimed that the single-atom structure is MN₄, while the dual-atom structure is M₂N₅. However, according to EXAFS fitting results (Table S1) and EDS mapping (Fig. 2), oxygen is dispersed in the sample. The author should try to identify the coordination environment of SACs and DACs.

Response:

We greatly appreciate your insightful comments. Indeed, as what you have mentioned, oxygen is dispersed in the sample. The possible reason for the dispersive of oxygen is that abundant oxygen-containing group are presented in the hollow carbon spheres, which are evidenced by the O 1s XPS spectra (Figure S46 in our revised supporting information). Besides, in order to identify the coordination environment of Co_{SA}-N-HCS-900 and Co₂-N-HCS-900, we have conducted the DFT calculations to investigate the possible structures containing the single Co atom structure and married Co₂ structure (model 1 to 10, Figure S34), and added the comparison between the simulated EXAFS and XANES spectra of the possible structures and the experimental spectra. It is found that the simulated spectra based on the single atom CoN₄

structure (model 9) agree well with the experimental EXAFS and XANES results of $\text{Co}_{\text{SA}}\text{-N-HCS-900}$, indicating that this model 9 is the most likely actual structure in $\text{Co}_{\text{SA}}\text{-N-HCS-900}$ (Figure S35, S36 and Table S2). As for $\text{Co}_2\text{-N-HCS-900}$, the simulated spectra based on the model 2 (binuclear Co_2N_5 configurations with oxygen) agree well with the experimental EXAFS and XANES results, indicating that this model is the most likely actual structure in $\text{Co}_2\text{-N-HCS-900}$, again confirming the formation of married $\text{Co}_2\text{N}_5\text{-OH}$ structure in $\text{Co}_2\text{-N-HCS-900}$ (Figure S37, S38 and Table S3). These results demonstrate that there are differences between Co single-atom of $\text{Co}_{\text{SA}}\text{-N-HCS-900}$ and Co dual-atom of $\text{Co}_2\text{-N-HCS-900}$.

To make it clear, the related part is listed as follows and highlighted in yellow.

“As displayed in Figure S46, the deconvolution of the O 1s spectra demonstrated the coexistence of oxygen-containing functional groups (C=O at ca. 531.8 eV, COOH at ca. 533.3 eV and absorbed water at ca. 536.3 eV) in $\text{Co}_{\text{NP}}\text{-HCS-900}$, $\text{Co}_{\text{SA}}\text{-N-HCS-900}$ and $\text{Co}_2\text{-N-HCS-900}$.”

Figure S46. O 1s high-resolution XPS spectra of (a) $\text{Co}_{\text{NP}}\text{-HCS-900}$, (b) $\text{Co}_{\text{SA}}\text{-N-HCS-900}$, and (c) $\text{Co}_2\text{-N-HCS-900}$.

“To further verify the possible structure of $\text{Co}_{\text{SA}}\text{-N-HCS-900}$, and $\text{Co}_2\text{-N-HCS-900}$, DFT calculations were first to investigate the possible structures containing the single Co atom structure and married Co_2 structure (model 1 to 10, Figure S34). Moreover, the comparison between the simulated EXAFS and XANES spectra of the possible structures and the experimental spectra were also recorded. As shown in Figure S35, S36 and Table S2, the simulated spectra based on the single atom CoN_4 model (model 9) agree well with the experimental EXAFS and XANES results of $\text{Co}_{\text{SA}}\text{-N-HCS-900}$, confirming that this model is the most likely actual structure in $\text{Co}_{\text{SA}}\text{-N-HCS-900}$. As for $\text{Co}_2\text{-N-HCS-900}$, the simulated spectra based on the model 2 (binuclear Co_2N_5 configurations with oxygen) agree well with

the experimental EXAFS and XANES results (Figure S37, S38 and Table S3), indicating that this model 2 is the most likely actual structure. These results synergistically validate that CoN_4 single-atom could couple with each other to form married Co_2N_5 dual-atom dimers via thermal migration process.”

Figure S34. The possible Co atomic structures optimized by DFT (green ball: Co; blue ball: N; purple ball: C; red ball: O)

Figure S35. Comparison between the Co K-edge XANES experimental spectrum of $\text{Co}_{\text{SA}}\text{-N-HCS-900}$ (solid red line) and the theoretical spectrum (solid blue line) calculated with different Co atomic structures (green ball: Co; blue ball: N; purple ball: C; red ball: O).

Figure S36. Comparison between the Co K-edge EXAFS experimental spectrum of Co_{SA}-N-HCS-900 (solid red line) and the theoretical spectrum (solid black line) calculated with different Co atomic structures (green ball: Co; blue ball: N; purple ball: C; red ball: O).

Figure S37. Comparison between the Co K-edge XANES experimental spectrum of Co₂-N-HCS-900 (solid red line) and the theoretical spectrum (solid blue line) calculated with different Co atomic structures (green ball: Co; blue ball: N; purple ball: C; red ball: O).

Figure S38. Comparison between the Co K-edge EXAFS experimental spectrum of Co₂-N-HCS-900 (solid red line) and the theoretical spectrum (solid black line) calculated with different Co atomic structures.

Table S2. The XANES and EXAFS fitting parameters at the Co K-edge of the possible structures from model 1 to model 10 for the Co_{SA}-N-HCS-900.

Model	XANES (S_0^2)	EXAFS (R-factor)
1	10.73	0.1370
2	3.42	0.0151
3	5.34	0.0403
4	12.76	0.1071

5	2.40	0.0737
6	1.51	0.1543
7	7.93	0.0195
8	6.39	0.0192
9	0.46	0.0068
10	4.68	0.0108

Table S3. The XANES and EXAFS fitting parameters at the Co K-edge of the possible structures from model 1 to model 10 for the Co₂-N-HCS-900.

Model	XANES (S_0^2)	EXAFS (R-factor)
1	13.50	0.1603
2	0.39	0.0033
3	6.34	0.0441
4	13.07	0.1184
5	3.31	0.0810
6	2.42	0.1825
7	7.73	0.0267
8	7.30	0.0254
9	3.58	0.0389
10	5.20	0.0128

(Please find the details in Page 12 and 14 of our revised manuscript and Page 39-43 ,51, 72 and 73 of our revised supporting information.)

Question 2:

The initial state of SACs in the calculation (Fig.1b), two metal atoms are adjacent. However, in the sample, the metal atoms are randomly dispersed. The author should prove that the metal atoms, which are separated by several carbon atoms, will also become dual-atom structures.

Response:

We greatly appreciate your insightful comments. According to your valuable suggestions, we have conducted the additional DFT calculations to prove that the metal atoms, which are

separated by several random carbon atoms (including six systems), will also become dual-atom structures (Figure S1). It can be seen that all the transforming processes, from two randomly dispersed CoN_4 single-atom separated by several carbon atoms to edge-adjacent Co_2N_6 and then to Co_2N_5 dual-atom, were exothermic in these additional six systems, again demonstrating that randomly dispersed CoN_4 single-atom would spontaneously be sintered to be dual-atom structure through thermal migration process.

To make it clear, the related part is listed as follows and highlighted in yellow.

Figure S1. (a-f) Calculated relative energies of different CoN_4 models that are separated by several carbon atoms, Co_2N_6 , and Co_2N_5 models.

“Figure 1b displays that the transforming processes, from neighbored two CoN_4 single-atom to edge-adjacent Co_2N_6 and then to Co_2N_5 dual-atom, were exothermic with the energy of 2.44 eV and 3.57 eV, respectively. Again, the processes from two randomly dispersed CoN_4 single-atom separated by several carbon atoms to edge-adjacent Co_2N_6 and then to Co_2N_5 dual-atom were exothermic (Figure S1), further decoding that randomly dispersed CoN_4 single-atom would spontaneously be sintered through thermal migration process.”

(Please find the details in Page 4 of our revised manuscript and Page 6 of our revised supporting information.)

Question 3:

The actual metal loading amount should be measured, including nanoparticles, single-atom catalysts, and dual-atom catalysts. Therefore, the mass activity of HER, OER,

and ORR should be compared as well.

Response:

We greatly appreciate your insightful comments. According to your valuable suggestions, we have conducted the inductivity coupled plasma optical emission spectrometry (ICP-OES) to determine the actual Co loading amount in the catalysts. The Co content is determined to be 1.48 wt.% for C_{ONP} -HCS-900, 1.41 wt.% for C_{OSA} -N-HCS-900, and 1.74 wt.% for Co_2 -N-HCS-900, respectively. Based on the ICP-OES results, the mass activity of HER, OER, and ORR were also calculated and compared as well. As shown in Figure S53, S55 and S56, the Co_2 -N-HCS-900 affords a higher mass activity of $97.6 \text{ A g}^{-1}_{Co}$ @ 0.9 V toward ORR, $2.79 \text{ A g}^{-1}_{Co}$ @ $\eta = 400 \text{ mV}$ toward OER, $1.31 \text{ A g}^{-1}_{Co}$ @ $\eta = 200 \text{ mV}$ toward HER than the control C_{ONP} -HCS-900, C_{OSA} -N-HCS-900 and commercial Pt/C, evidencing that more abundant accessible active sites are created toward catalyzing ORR, OER and HER.

To make it clear, the related part is listed as follows and highlighted in yellow.

“The Co content, measured by the inductivity coupled plasma optical emission spectrometry (ICP-OES), is determined to be 1.48 wt.% for C_{ONP} -HCS-900, 1.41 wt.% for the C_{OSA} -N-HCS-900 and 1.74 wt.% for the Co_2 -N-HCS-900, respectively. Taking into account the BET surface area and Co content, the Co_2 -N-HCS-900 affords a higher atomic Co coverage of 0.233 atoms per square nanometer than the C_{OSA} -N-HCS-900 (0.196 atoms per square nanometer), decoding more accessible active Co sites created on the former.”

“Moreover, Co_2 -N-HCS-900 affords a higher electrochemical double-layer capacitance (C_{dl}) value of 195.1 mF cm^{-2} and mass activity of $97.6 \text{ A g}^{-1}_{Co}$ @ 0.9 V than the control catalysts (C_{ONP} -HCS-900, C_{OSA} -N-HCS-900 and commercial Pt/C), evidencing that more abundant accessible active sites are created toward catalyzing ORR (Figure S51-S53).”

“Also, the Co_2 -N-HCS-900 pays a large mass activity of $2.79 \text{ A g}^{-1}_{Co}$ @ $\eta = 400 \text{ mV}$ toward OER and $1.31 \text{ A g}^{-1}_{Co}$ @ $\eta = 200 \text{ mV}$ toward HER than the control C_{ONP} -HCS-900, C_{OSA} -N-HCS-900 and commercial Pt/C (Figure S55 and S56).”

Figure S53. Mass activity toward ORR for the $\text{Co}_{\text{NP}}\text{-HCS-900}$, $\text{Co}_{\text{SA}}\text{-N-HCS-900}$, $\text{Co}_2\text{-N-HCS-900}$ and commercial Pt/C.

Figure S55. Mass activity toward OER for the $\text{Co}_{\text{NP}}\text{-HCS-900}$, $\text{Co}_{\text{SA}}\text{-N-HCS-900}$, $\text{Co}_2\text{-N-HCS-900}$ and commercial RuO_2 .

Figure S56. Mass activity toward HER for the $\text{Co}_{\text{NP}}\text{-HCS-900}$, $\text{Co}_{\text{SA}}\text{-N-HCS-900}$, $\text{Co}_2\text{-N-HCS-900}$ and commercial Pt/C.

(Please find the details in Page 14-17 of our revised manuscript and Page 58 and 60-61 of our revised supporting information.)

We really appreciate your valuable and constructive suggestions, and we hope you are satisfied with the present manuscript which is obviously improved with the help from you.

Reviewers' comments:

Reviewer #1 (Remarks to the Author):

From my side, the authors didn't address the proposed questions directly. The quality of this updated version isn't largely improved, and still can't meet the standard of Nature Communications.

1. The authors attempted to establish the brand new correlation between spin states and intermediates adsorption/desorption. However, only three samples couldn't represent the inevitability of this correlation. In addition, recently published papers pointed that higher spin state is beneficial to the triplet oxygen molecule evolution (Angew. Chem. Int. Ed. 2020, 59, 2313; Chem 2017, 3, 812; Adv. Mater. 2018, 30, 1803220; Adv. Mater. 2020, 32, 1907976), but the authors recognize lower spin state is good for reaction. Please replenish more data to support the viewpoint directly.
2. Based on Sabatier principle, appropriate energy of intermediates adsorption/desorption is important, both too strong and too weak are not suitable for reaction. So why do the authors recognize the energy of intermediates adsorption/desorption for Co₂N₅ the most appropriate? Thus, it's indispensable to expand the sample size.
3. From the figure S25, only a few hollow spheres contain Co particles. Based on the TEM images, it should be the mixture of Co particles and carbon-based material. The name of sample is inappropriate.
4. The authors employed EPR measurement to investigate the magnetism of samples. However, the authors didn't analyse it distinctly. The summary of EPR measurement failed to disclose the information of spin states. This does not combine experiments well with DFT calculations.
5. The oxidation peaks of cobalt could influence the assessment of the true OER performance. Please add the reverse sweep LSV curves of samples.

Reviewer #2 (Remarks to the Author):

The author has provided detailed responses to most of the questions, but the scientific significance of this manuscript can't meet the stringent standard of Nature Communications at current stage. Some additional comments are as follows.

1. There is a lack of positive responses to individual questions such as "Why Co₁₀ is taken as nanoparticles sample for research?" and "The common conclusion of previously reported researches is that the increase of magnetic moment is conducive to the improvement of catalytic activity. However, the relationship between the size of magnetic moment and the level of activity in this work is inconsistent with it. Please make corresponding comparisons and explain accordingly."
2. In addition, the author mentioned "it was found that the spin state of Co atoms can be harmonized from NP-to-SA-to-DA, in which the Co₂N₅ dual-atom with low spin-state can achieve the ideal balanced adsorption/desorption of O* and H* intermediates." in the article, but no clear evidence was provided in the experiment. EPR is only qualitative analysis and cannot prove the high/low spin states of the metal Co atom.

Reviewer #3 (Remarks to the Author):

The authors addressed the raised concerned. This reviewer can now recommend the publication of the manuscript.

Reviewer #1 (Remarks to the Author):

From my side, the authors didn't address the proposed questions directly. The quality of this updated version isn't largely improved, and still can't meet the standard of Nature Communications.

Response:

We greatly appreciate your critical comments. Keeping the reviewer's suggestions in mind, we now obtained more evidence to address the proposed questions. Moreover, we have responded point by point to the reviewer's comment as listed in this letter. The changes have been highlighted with yellow in the revised manuscript and supporting information.

Question 1:

The authors attempted to establish the brand new correlation between spin states and intermediates adsorption/desorption. However, only three samples couldn't represent the inevitability of this correlation. In addition, recently published papers pointed that higher spin state is beneficial to the triplet oxygen molecule evolution (Angew. Chem. Int. Ed. 2020, 59, 2313; Chem 2017, 3, 812; Adv. Mater. 2018, 30, 1803220; Adv. Mater. 2020, 32, 1907976), but the authors recognize lower spin state is good for reaction. Please replenish more data to support the viewpoint directly.

Response:

We greatly appreciate your insightful comments. According to your valuable suggestions, we have added more samples (Co₁₆ and Co₄ nanoparticles models) to establish the brand correlation between spin states and intermediates adsorption/desorption. As shown in Figure 1e, g and S18, it can be clearly captured that the spin magnetic moment presents a positive and quasi-linear correlation ($R^2=0.962$) with the $|\Delta G_{\text{OOH}^*-\text{OH}^*}|$, in the positive sequence from Co₂N₅ dual-atom (0.05 μ_B , 2.55 eV) to CoN₄ single-atom (0.81 μ_B , 3.12 eV) to Co₄ nanoparticles (1.28 μ_B , 3.46 eV) to Co₁₀ nanoparticles (1.48 μ_B , 3.79 eV) and to Co₁₆ nanoparticles (1.87 μ_B , 4.36 eV). As for HER, we can also observe a negative and quasi-linear correlation between the spin

magnetic moment and the Gibbs free energy of H* (ΔG_{H^*}), in which the ΔG_{H^*} decreases in a linear manner ($R^2=0.959$) with the upshift of the spin magnetic moment (Figure 1g). The Co₂N₅ dual-atom possesses the highest ΔG_{H^*} of -0.04 eV (close to the ideal value of 0 eV) among the three investigated models, validating that the low spin state enables the Co₂N₅ dual-atom with moderate adsorption and desorption of H* intermediate.

Indeed, as what you have mentioned, recently published papers pointed that higher spin state is beneficial to the triplet oxygen molecule evolution (*Chem*, **2017**, 3, 812; *Adv. Mater.*, **2018**, 30, 1803220; *Adv. Mater.*, **2020**, 32, 1907976). In the literatures you mentioned (*Angew. Chem. Int. Ed.*, **2020**, 59, 2313), the authors point that the too high spin state could result in the suppression the bonding strength with reaction intermediate, and concluded that the strong interaction between Fe^{III} and TiO₂ can lower the spin-state of Fe^{III} and enhance the conductivity of catalyst with less ohmic loss for better OER activity. In our manuscript, we recognize lower spin state is good for reaction, which is similar to the conclusion from the previous literatures ((1) *Energy Environ Sci.*, **2016**, 9, 2418-2432. (2) *Adv. Funct. Mater.*, **2019**, 29, 1906174. (3) *Appl. Catal. B: Environ.*, **2023**, 323, 122163. (4) *Angew. Chem. Int. Ed.*, **2022**, 61, 202114293. (5) *Angew. Chem. Int. Ed.*, **2022**, 61, 202201007. (6) *Angew. Chem. Int. Ed.*, **2021**, 133, 25608-25614. (7) *Appl. Catal. B: Environ.*, **2021**, 285, 119778). Besides, according to your kind suggestions, we have added more data to support the viewpoint directly. To disclose the correlation between spin configuration and free energy, the crystal orbital Hamilton population (COHP) is calculated to compare the bonding character of absorbed O* onto Co₁₀, CoN₄ and Co₂N₅ models. As is well known, the positive/negative COHP was on behalf of the bonding/antibonding states, respectively. The Co-O bonding strength can be evaluated by the integrated-COHP (-ICOHP) values, which could quantitatively describe the d-p hybridization strength. It can be seen in Figure S20 that the -ICOHP value was determined to be -0.22 for Co₁₀ model, -0.47 for CoN₄, and -0.45 for Co₂N₅ model, respectively. The middle -ICOHP value between three investigated models evidently validate the moderated Co-O affinity in the Co₂N₅ model, decoding that it could achieve the optimized O* adsorption/desorption, thus

boosting the ORR/OER activities. As shown in Figure S21, a stronger antibonding state for the Co₂N₅ model appear at Fermi level among three investigated models, suggesting that the more electrons transfer from Co-3d orbital to the unfilled O-2p orbital, thus leading to a lower reaction activation energy and the enhanced conductivity of catalyst with less ohmic loss for better catalytic activity ((1) *Nat. Commun.*, **2021**, 12, 4827. (2) *Angew. Chem. Int. Ed.*, **2020**, 59, 2313).

To make it clear, the related part is listed as follows and highlighted in yellow.

Next, we investigated the electronic structure of Co₁₆ nanoparticles, Co₁₀ nanoparticles, Co₄ nanoparticles, CoN₄ single-atom, and Co₂N₅ dual-atom models (Figure S17). The charge density difference and Bader charge results in Figure 1c validate that the Co atoms of Co₂N₅ dual-atom possess a higher charge of 1.11e than that of CoN₄ single-atom (1.03e) and Co₁₀ nanoparticles (0.11e). As shown in the projected density of states (PDOS) diagram (Figure S18), the electrons of Co-3d orbitals are asymmetrically arranged in the spin channels for Co₁₆ nanoparticles, Co₁₀ nanoparticles, Co₄ nanoparticles, CoN₄ single-atom, and Co₂N₅ dual-atom, showing the magnetic moment of 1.87 μ_B , 1.48 μ_B , 1.28 μ_B , 0.81 μ_B , and 0.05 μ_B , respectively. Such decreased spin magnetic moment mainly results from the redistribution of electrons of Co-3d orbital triggered by the energy levels rearrangement, which would give rise to the increased filling degree of d_{z^2} orbitals and induce the weakened adsorption of reaction intermediates (OOH*, O* and OH* toward ORR/OER, H* toward HER), thus boosting the catalytic activities. Further support for this phenomenon can be demonstrated by the downshift of d band center of Co-3d orbitals from Co₁₆ nanoparticles to Co₁₀ nanoparticles to Co₄ nanoparticles to CoN₄ single-atom, to Co₂N₅ dual-atom, accompanied with the corresponding value of -0.52, -0.71, -1.27, -1.61 and -2.27 eV (Figure 1d).

We also investigated the Gibbs free energy toward ORR/OER/HER of Co₁₀ nanoparticles, CoN₄ single-atom, and Co₂N₅ dual-atom models to afford a deeper understanding of the catalytic mechanism and origin of superior catalytic activities. It is well known that the Gibbs free energy difference between ΔG_{OOH^*} and ΔG_{OH^*} ($|\Delta G_{\text{OOH}^*-\text{OH}^*}|$) could serve as the important ORR/OER reaction descriptor, with an ideal

value of 2.46 eV.³⁸⁻⁴⁰ The $|\Delta G_{\text{OOH}^*-\text{OH}^*}|$ is significantly restricted by the adsorption affinity of O^* , in which either too strong or too weak adsorption of O^* could result in the increased value of $|\Delta G_{\text{OOH}^*-\text{OH}^*}|$.⁴¹ As shown in Figure 1e, g, S18, it can be clearly captured that the spin magnetic moment presents a positive and quasi-linear correlation ($R^2=0.962$) with the $|\Delta G_{\text{OOH}^*-\text{OH}^*}|$, in the positive sequence from Co_2N_5 dual-atom ($0.05 \mu_{\text{B}}$, 2.55 eV) to CoN_4 single-atom ($0.81 \mu_{\text{B}}$, 3.12 eV) to Co_4 nanoparticles ($1.28 \mu_{\text{B}}$, 3.46 eV) to Co_{10} nanoparticles ($1.48 \mu_{\text{B}}$, 3.79 eV) and to Co_{16} nanoparticles ($1.87 \mu_{\text{B}}$, 4.36 eV). A universal ΔG_{OOH^*} to ΔG_{OH^*} scaling relation with an average $|\Delta G_{\text{OOH}^*-\text{OH}^*}|$ value of 3.2 eV has been established for most conventional catalysts, whereas the ideal ΔG_{OOH^*} to ΔG_{OH^*} scaling relation possesses an average $|\Delta G_{\text{OOH}^*-\text{OH}^*}|$ value of 2.46 eV for ideal catalysts (Figure 1f).^{38, 39} The ΔG_{OOH^*} to ΔG_{OH^*} coordinate point of Co_2N_5 dual-atom is located at the ideal scaling relations accompanied with the $|\Delta G_{\text{OOH}^*-\text{OH}^*}|$ value of 2.55 eV, which is very close to the 2.46 eV for ideal catalysts toward ORR/OER. This decodes that the energy of O^* intermediates adsorption/desorption for Co_2N_5 is the most appropriate. The possible reason for this phenomenon is that the Co_2N_5 dual-atom with the decreased spin magnetic moment can break the universal ΔG_{OOH^*} to ΔG_{OH^*} scaling relation, thus achieving the ideal balanced O^* adsorption. Such phenomenon can be further supported by the moderate O^* adsorption energy (1.89 eV, Figure S19) of Co_2N_5 dual-atom model among investigated models, decoding that it could achieve the optimized O^* adsorption/desorption, thus boosting the ORR/OER activities. To disclose the correlation between spin configuration and free energy of O^* , the crystal orbital Hamilton population (COHP) is calculated to compare the bonding character of absorbed O^* onto Co_{10} , CoN_4 and Co_2N_5 models. As is well known, the positive/negative COHP was on behalf of the bonding/antibonding states, respectively. The Co-O bonding strength can be evaluated by the integrated-COHP (-ICOHP) values, which could quantitatively describe the d-p hybridization strength. It can be seen in Figure S20 that the -ICOHP value was determined to be -0.22 for Co_{10} model, -0.47 for CoN_4 , and -0.45 for Co_2N_5 model, respectively. The middle -ICOHP value between three investigated models evidently validating the moderated Co-O affinity in the Co_2N_5 model, decoding that it could achieve the optimized O^* adsorption/desorption,

thus boosting the ORR/OER activities. As shown in Figure S21, a stronger antibonding state for the Co_2N_5 model appear at Fermi level among three investigated models, suggesting that the more electrons from Co-3d orbital transfer to the unfilled O-2p orbital, thus leading to a lower reaction activation energy and the enhanced conductivity of catalyst with less ohmic loss for better catalytic activity.^{43,44} As for HER, we can also observe a negative and quasi-linear correlation between the spin magnetic moment and the Gibbs free energy of H^* (ΔG_{H^*}), in which the ΔG_{H^*} decreases in a linear manner ($R^2=0.959$) with the upshift of the spin magnetic moment (Figure 1g). The Co_2N_5 dual-atom possesses the highest ΔG_{H^*} of -0.04 eV (close to the ideal value of 0 eV) among the three investigated models, validating that the low spin state enables the Co_2N_5 dual-atom with moderate adsorption and desorption of H^* intermediate. Besides, the highest H^* adsorption energy of -0.21 eV can be captured among the three investigated models (Figure S22), again validating that it affords moderate adsorption and desorption of H^* intermediate, thus endowing excellent HER activities.

Figure 1. (a) Calculated relative energies along the stretching pathway of the Co atom from Co₁₀ to Co_{SA}/C or CoN₄ model by CI-NEB, (b) Calculated relative energies of CoN₄, Co₂N₆, and Co₂N₅ models. (c) The charge density difference and Bader charge diagrams, (d) PDOS, (e) linear correlation between magnetic moment and $|\Delta G_{\text{OOH}^*-\text{OH}^*}|$, (f) the ΔG_{OOH^*} to ΔG_{OH^*} scaling for Co₁₆, Co₁₀, Co₄, CoN₄, and Co₂N₅ models relative to the universal and ideal scaling lines, and (g) linear correlation between the magnetic moment and ΔG_{H^*} . Free energy diagrams of Co-N₄ and Co₂N₅ models for (h) ORR, (i) OER, and (j) HER.

Figure S18. PDOS of the Co_{16} , Co_{10} , Co_4 , CoN_4 , and Co_2N_5 models.

Figure S20. Crystal orbital Hamilton population (COHP) analysis corresponding integrated-COHP (-ICOHP) value of the $\text{Co}_{10}\text{-O}$ (a), $\text{CoN}_4\text{-O}$ (b) and $\text{Co}_2\text{N}_5\text{-O}$ (c).

Figure S21. The COHP analysis (a) and the magnification around Fermi level image (b) of the $\text{Co}_{10}\text{-O}$, $\text{CoN}_4\text{-O}$ and $\text{Co}_2\text{N}_5\text{-O}$.

(Please find the details in Page 5-8 of our revised manuscript and Page 22-27 of our revised supporting information.)

Question 2:

Based on Sabatier principle, appropriate energy of intermediates adsorption/desorption is important, both too strong and too weak are not suitable for reaction. So why do the authors recognize the energy of intermediates adsorption/desorption for Co_2N_5 the most appropriate? Thus, it's indispensable to expand the sample size.

Response:

We greatly appreciate your insightful comments. Indeed, as what you mentioned, appropriate energy of intermediates adsorption/desorption is important based on the Sabatier principle. Both too strong and too weak are not suitable for reaction. As for ORR/OER, it is well known that the Gibbs free energy difference between ΔG_{OOH^*} and ΔG_{OH^*} ($|\Delta G_{\text{OOH}^*-\text{OH}^*}|$) could serve as the important ORR/OER reaction descriptor, with an ideal value of 2.46 eV. **The $|\Delta G_{\text{OOH}^*-\text{OH}^*}|$ is significantly restricted by the adsorption affinity of O^* , in which either too strong or too weak adsorption of O^* could result in the increased value of $|\Delta G_{\text{OOH}^*-\text{OH}^*}|$.** ((1) *Angew. Chem. Int. Ed.*, **2022**, *134*, 202202200. (2) *Nano Lett.*, **2022**, *22*, 3392-3399. (3) *Angew. Chem. Int. Ed.*, **2022**, *134*, 202211098. (4) *Adv. Mater.*, **2023**, 2300381. (5) *Energy Environ. Sci.*, **2022**, *15*, 2619-2628. (6) *Adv. Funct. Mater.*, **2023**, 2301559. (7) *Adv. Mater.*, **2021**, *33*, 2102595. (8) *Adv. Funct. Mater.*, **2021**, *31*, 2101239. (9) *J. Energy Chem.*, **2021**, *55*, 162-168. (10) *Adv. Energy Mater.*, **2023**, *13*, 2203159.) Based on these literatures, a universal ΔG_{OOH^*} to ΔG_{OH^*} scaling relation with an average $|\Delta G_{\text{OOH}^*-\text{OH}^*}|$ value of 3.2 eV has been established for most conventional catalysts, whereas the ideal ΔG_{OOH^*} to ΔG_{OH^*} scaling relation possesses an average $|\Delta G_{\text{OOH}^*-\text{OH}^*}|$ value of 2.46 eV for ideal catalysts (Figure 1f). In our manuscript, the ΔG_{OOH^*} to ΔG_{OH^*} coordinate point of Co_2N_5 dual-atom is located at the ideal scaling relations accompanied with the $|\Delta G_{\text{OOH}^*-\text{OH}^*}|$ value of 2.55 eV, which is very close to the 2.46 eV for ideal catalysts. This decodes that the energy of O^* intermediates adsorption/desorption for Co_2N_5 is the most appropriate. Therefore, we recognize that the energy of O^* intermediates adsorption/desorption for Co_2N_5 is the most appropriate.

Besides, according to your kind suggestions, we have added more samples (Co₄ and Co₁₆ nanoparticles models) to establish the brand correlation between spin states and intermediates adsorption/desorption. As shown in Figure 1e, g and S18, it can be clearly captured that the spin magnetic moment presents a positive and quasi-linear correlation ($R^2=0.962$) with the $|\Delta G_{\text{OOH}^*-\text{OH}^*}|$, in the positive sequence from Co₂N₅ dual-atom (0.05 μ_B , 2.55 eV) to CoN₄ single-atom (0.81 μ_B , 3.12 eV) to Co₄ nanoparticles (1.28 μ_B , 3.46 eV) to Co₁₀ nanoparticles (1.48 μ_B , 3.79 eV) and to Co₁₆ nanoparticles (1.87 μ_B , 4.36 eV). It can be seen the Co₂N₅ dual-atom affords the $|\Delta G_{\text{OOH}^*-\text{OH}^*}|$ value of 2.55 eV, which is very close to the 2.46 eV for ideal catalysts, decoding that the energy of O* intermediates adsorption/desorption for Co₂N₅ is the most appropriate. As for HER, we can also observe a negative and quasi-linear correlation between the spin magnetic moment and the Gibbs free energy of H* (ΔG_{H^*}), in which the ΔG_{H^*} decreases in a linear manner ($R^2=0.959$) with the upshift of the spin magnetic moment (Figure 1g). The Co₂N₅ dual-atom possesses the highest ΔG_{H^*} of -0.04 eV (close to the ideal value of 0 eV) among the investigated models, validating that the low spin state enables the Co₂N₅ dual-atom with moderate adsorption and desorption of H* intermediate.

To make it clear, the related part is listed as follows and highlighted in yellow.

Next, we investigated the electronic structure of Co₁₆ nanoparticles, Co₁₀ nanoparticles, Co₄ nanoparticles, CoN₄ single-atom, and Co₂N₅ dual-atom models (Figure S17). The charge density difference and Bader charge results in Figure 1c validate that the Co atoms of Co₂N₅ dual-atom possess a higher charge of 1.11e than that of CoN₄ single-atom (1.03e) and Co₁₀ nanoparticles (0.11e). As shown in the projected density of states (PDOS) diagram (Figure S18), the electrons of Co-3d orbitals are asymmetrically arranged in the spin channels for Co₁₆ nanoparticles, Co₁₀ nanoparticles, Co₄ nanoparticles, CoN₄ single-atom, and Co₂N₅ dual-atom, showing the magnetic moment of 1.87 μ_B , 1.48 μ_B , 1.28 μ_B , 0.81 μ_B , and 0.05 μ_B , respectively. Such decreased spin magnetic moment mainly results from the redistribution of electrons of Co-3d orbital triggered by the energy levels rearrangement, which would give rise to the increased filling degree of d_{z^2} orbitals and induce the weakened adsorption of

reaction intermediates (OOH*, O* and OH* toward ORR/OER, H* toward HER), thus boosting the catalytic activities. Further support for this phenomenon can be demonstrated by the downshift of d band center of Co-3d orbitals from Co₁₆ nanoparticles to Co₁₀ nanoparticles to Co₄ nanoparticles to CoN₄ single-atom, to Co₂N₅ dual-atom, accompanied with the corresponding value of -0.52, -0.71, -1.27, -1.61 and -2.27 eV (Figure 1d).

We also investigated the Gibbs free energy toward ORR/OER/HER of Co₁₀ nanoparticles, CoN₄ single-atom, and Co₂N₅ dual-atom models to afford a deeper understanding of the catalytic mechanism and origin of superior catalytic activities. It is well known that the Gibbs free energy difference between ΔG_{OOH^*} and ΔG_{OH^*} ($|\Delta G_{\text{OOH}^*-\text{OH}^*}|$) could serve as the important ORR/OER reaction descriptor, with an ideal value of 2.46 eV.³⁸⁻⁴⁰ The $|\Delta G_{\text{OOH}^*-\text{OH}^*}|$ is significantly restricted by the adsorption affinity of O*, in which either too strong or too weak adsorption of O* could result in the increased value of $|\Delta G_{\text{OOH}^*-\text{OH}^*}|$.⁴¹ As shown in Figure 1e, g, S18, it can be clearly captured that the spin magnetic moment presents a positive and quasi-linear correlation ($R^2=0.962$) with the $|\Delta G_{\text{OOH}^*-\text{OH}^*}|$, in the positive sequence from Co₂N₅ dual-atom (0.05 μ_B , 2.55 eV) to CoN₄ single-atom (0.81 μ_B , 3.12 eV) to Co₄ nanoparticles (1.28 μ_B , 3.46 eV) to Co₁₀ nanoparticles (1.48 μ_B , 3.79 eV) and to Co₁₆ nanoparticles (1.87 μ_B , 4.36 eV). A universal ΔG_{OOH^*} to ΔG_{OH^*} scaling relation with an average $|\Delta G_{\text{OOH}^*-\text{OH}^*}|$ value of 3.2 eV has been established for most conventional catalysts, whereas the ideal ΔG_{OOH^*} to ΔG_{OH^*} scaling relation possesses an average $|\Delta G_{\text{OOH}^*-\text{OH}^*}|$ value of 2.46 eV for ideal catalysts (Figure 1f).^{38,39} The ΔG_{OOH^*} to ΔG_{OH^*} coordinate point of Co₂N₅ dual-atom is located at the ideal scaling relations accompanied with the $|\Delta G_{\text{OOH}^*-\text{OH}^*}|$ value of 2.55 eV, which is very close to the 2.46 eV for ideal catalysts toward ORR/OER. This decodes that the energy of O* intermediates adsorption/desorption for Co₂N₅ is the most appropriate. The possible reason for this phenomenon is that the Co₂N₅ dual-atom with the decreased spin magnetic moment can break the universal ΔG_{OOH^*} to ΔG_{OH^*} scaling relation, thus achieving the ideal balanced O* adsorption. Such phenomenon can be further supported by the moderate O* adsorption energy (1.89 eV, Figure S19) of Co₂N₅ dual-atom model among investigated models, decoding that it

could achieve the optimized O^* adsorption/desorption, thus boosting the ORR/OER activities.

Figure 1. (a) Calculated relative energies along the stretching pathway of the Co atom from Co₁₀ to Co_{SA}/C or CoN₄ model by CI-NEB, (b) Calculated relative energies of CoN₄, Co₂N₆, and Co₂N₅ models. (c) The charge density difference and Bader charge diagrams, (d) PDOS, (e) linear correlation between magnetic moment and $|\Delta G_{OOH^*OH^*}|$, (f) the ΔG_{OOH^*} to ΔG_{OH^*} scaling for Co₁₆, Co₁₀, Co₄, CoN₄, and Co₂N₅ models relative to the universal and ideal scaling lines, and (g) linear correlation between the magnetic moment and ΔG_{H^*} . Free energy diagrams of Co-N₄ and Co₂N₅ models for (h) ORR, (i) OER, and (j) HER.

Figure S18. PDOS of the Co₁₆, Co₁₀, Co₄, CoN₄, and Co₂N₅ models.

(Please find the details in Page 5-8 of our revised manuscript and Page 22-27 of our revised supporting information.)

Question 3:

From the figure S25, only a few hollow spheres contain Co partials. Based on the TEM images, it should be the mixture of Co partials and carbon-based material. The name of sample is inappropriate.

Response:

We greatly appreciate your insightful comments. According to your valuable suggestions, we have corrected the name of sample (Co_{NP}-HCS-900) to Co_{NP}/HCS-900 in our revised manuscript and supporting information.

To make it clear, the related part is listed as follows and highlighted in yellow.

“Motivated by DFT analysis, a series of catalysts were prepared, involving Co_{NP}/HCS-900 with aggregated Co NPs, Co_{SA}-N-HCS-900 with CoN₄ single-atom, and Co₂-N-HCS-900 with married Co₂N₅ dual-atom (Figure 2a). In detail, a facile double-solvent impregnation method was adopted to prepare the Co-hollow polymer spheres (Co-HPS) precursor. Then Co_{NP}/HCS-900 was obtained by directly pyrolyzing the Co-HPS, and the Co_{NP}/N-HCS-300 are synthesized by pyrolyzing the Co-HPS with

melamine at 300 °C. Besides, Co_{SA}-N-HCS-900 can be synthesized when the melamine was added in the above pyrolysis process since the melamine can be decomposed at high temperature (>400 °C) and serve as the N agent that is generally coordinated with Co atoms to form Co-N_x moieties for promoting the Co nanoparticles atomization

“The ORR/OER/HER performance for the obtained catalysts was first investigated (Figure 5a-f). Linear sweep voltammetry (LSV) curves display that Co₂-N-HCS-900 owns a more positive onset potential of 0.99 V and half-wave potential of 0.86 V toward ORR than HCS-900, Co_{NP}/HCS-900, and Co_{SA}-N-HCS-900 (Figure 5a and S54), which are even comparable to that of commercial Pt/C and other reported non-precious M-N-C catalysts (Table S4). Its superior ORR kinetics can be again decoded by the smallest Tafel slope (48.0 mV dec⁻¹) and highest kinetic current density (8.33 mA cm⁻² @ 0.85V) as shown in Figure 5d and S55. Moreover, Co₂-N-HCS-900 affords a higher electrochemical double-layer capacitance (C_{dl}) value of 195.1 mF cm⁻² and mass activity of 97.6 A g⁻¹Co @ 0.9 V than the control catalysts (Co_{NP}/HCS-900, Co_{SA}-N-HCS-900 and commercial Pt/C), evidencing that more abundant accessible active sites are created toward catalyzing ORR (Figure S56-58)

Question 4:

The authors employed EPR measurement to investigate the magnetism of samples. However, the authors didn't analyze it distinctly. The summary of EPR measurement failed to disclose the information of spin states. This does not combine experiments well with DFT calculations.

Response:

We greatly appreciate your insightful comments. According to your valuable suggestions, we have performed the ferromagnetic hysteresis loop and zero-field cooling (ZFC) temperature-dependent magnetic susceptibility measurements to reveal the detailed information of spin states for the investigated catalysts. As shown in Figure 4g, the ferromagnetic hysteresis loop of investigated catalysts at 300 K exhibits a saturation magnetization. Noted that the saturation magnetization displays a downward trend from Co_{NP}/HCS-900 to Co_{SA}-N-HCS-900 to Co₂-N-HCS-900. An enlarged view

of the curve around $H=0$ decodes that the $\text{Co}_2\text{-N-HCS-900}$ affords a lowest coercive magnetic field and residual magnetization (inset of Figure 4g). In order to further reveal the electron spin state of catalysts, the ZFC temperature-dependent magnetic susceptibility was conducted. The unpaired d-band electrons (n) of catalysts can be calculated based on the following formula:

$$\bar{\mu}_{\text{eff}}(t) = \sum_i c_i(t) \mu_i^2 \quad \text{with} \quad \sum_i c_i(t) = 1 \quad (1)$$

$$\mu_{\text{eff}} = \sqrt{n(n+2)} \quad (2)$$

Here, μ_{eff} is the effective magnetic moment of the transition metal ions, c_i is the corresponding concentrations of Co, and n is the number of unpaired d electron of the Co-3d orbitals. As presented in Figure 4i and j, the effective magnetic moment of the $\text{Co}_{\text{SA}}\text{-N-HCS-900}$ and $\text{Co}_2\text{-N-HCS-900}$ can be calculated to be 2.6 and 1.7 μ_{eff} , respectively. The average number of the unpaired electron is 1.0 in Co-3d orbitals for the $\text{Co}_2\text{-N-HCS-900}$, which is lower than that of $\text{Co}_{\text{SA}}\text{-N-HCS-900}$ (1.7), indicating the decreased electron spin polarization from $\text{Co}_{\text{SA}}\text{-N-HCS-900}$ to $\text{Co}_2\text{-N-HCS-900}$. Based on these above-mentioned results, one may conclude that the spin magnetic moment displays a downward trend from the $\text{Co}_{\text{NP}}/\text{HCS-900}$ to $\text{Co}_{\text{SA}}\text{-N-HCS-900}$ to $\text{Co}_2\text{-N-HCS-900}$.

To make it clear, the related part is listed as follows and highlighted in yellow.

“As is well known, the X-band electron paramagnetic resonance (EPR) measurement is a powerful tool to investigate the paramagnetic properties of catalysts. It can be seen in Figure 4f that a strong signal can be detected for the $\text{Co}_{\text{NP}}/\text{HCS-900}$, indicating that it is paramagnetic. A downward trend can be captured from the $\text{Co}_{\text{NP}}/\text{HCS-900}$ to $\text{Co}_{\text{SA}}\text{-N-HCS-900}$ to $\text{Co}_2\text{-N-HCS-900}$, indicating that the spin magnetic moment was decreased from Co nanoparticles to single CoN_4 sites to married Co_2N_5 sites. Notably, no signal appears for the $\text{Co}_2\text{-N-HCS-900}$, possibly due to the formation of the binuclear Co structure with antiferromagnetic coupling sites, again confirming that the married Co_2N_5 structure can be successfully constructed in the $\text{Co}_2\text{-N-HCS-900}$. As shown in Figure 4g, the ferromagnetic hysteresis loop of investigated catalysts at 300 K exhibits a saturation magnetization. Noted that the saturation

magnetization displays a downward trend from $\text{Co}_{\text{NP}}/\text{HCS-900}$ to $\text{Co}_{\text{SA-N}}/\text{HCS-900}$ to $\text{Co}_{\text{O}_2\text{-N}}/\text{HCS-900}$. An enlarged view of the curve around $H=0$ decodes that the $\text{Co}_{\text{O}_2\text{-N}}/\text{HCS-900}$ affords a lowest coercive magnetic field and residual magnetization (inset of Figure 4g). In order to further reveal the electron spin configuration of the investigated catalysts, the zero-field cooling (ZFC) temperature-dependent magnetic susceptibility was conducted (Figure 4h and Figure S51). As presented in Figure 4h and i, the effective magnetic moment of the $\text{Co}_{\text{SA-N}}/\text{HCS-900}$ and $\text{Co}_{\text{O}_2\text{-N}}/\text{HCS-900}$ can be calculated to be 2.6 and 1.7 μ_{eff} , respectively. The average number of the unpaired electron is 1.0 in Co-3d orbitals for the $\text{Co}_{\text{O}_2\text{-N}}/\text{HCS-900}$, which is lower than that of $\text{Co}_{\text{SA-N}}/\text{HCS-900}$ (1.7), indicating the decreased electron spin polarization from $\text{Co}_{\text{SA-N}}/\text{HCS-900}$ to $\text{Co}_{\text{O}_2\text{-N}}/\text{HCS-900}$. Based on these above-mentioned results, one may conclude that the spin magnetic moment displays a downward trend from the $\text{Co}_{\text{NP}}/\text{HCS-900}$ to $\text{Co}_{\text{SA-N}}/\text{HCS-900}$ to $\text{Co}_{\text{O}_2\text{-N}}/\text{HCS-900}$. Ultraviolet photoemission spectroscopy (UPS) was conducted to investigate the electronic state of the $\text{Co}_{\text{NP}}/\text{HCS-900}$, $\text{Co}_{\text{SA-N}}/\text{HCS-900}$ and $\text{Co}_{\text{O}_2\text{-N}}/\text{HCS-900}$. The work function represents the minimum energy for inner electrons to escape from the surface of catalysts.....”

Figure 4. (a) XANES and (b) comparisons of the q-space magnitudes of Co foil, Co_{SA}-N-HCS-900 and Co₂-N-HCS-900. (c) WT-EXAFS of Co foil, CoO, CoPc, Co_{SA}-N-HCS-900, and Co₂-N-HCS-900 at Co K-edge. (d and e) The corresponding k^3 -weighted EXAFS fitting curves at R space for the Co_{SA}-N-HCS-900 and Co₂-N-HCS-900 (inset: schematic structure of Co-N₄ and Co₂N₅ model, (green: Co, blue: N, red:O)). (f) X-band electron paramagnetic resonance (EPR) spectrum (g) magnetic hysteresis loops at room temperature (300 K) and inset image of the magnification of magnetic hysteresis loops around H=0 for the Co_{NP}/HCS-900, Co_{SA}-N-HCS-900, and Co₂-N-HCS-900. (h) M-T curves and (i) the corresponding unpaired electron n and effective magnetic moment (μ_{eff}) of the Co_{SA}-N-HCS-900 and Co₂-N-HCS-900. (j) The work function spectrum for the Co_{NP}/HCS-900, Co_{SA}-N-HCS-900, and Co₂-N-HCS-900.

Figure S51. M-T curve of the Co_{NP}/HCS-900.

(Please find the details in Page14-16 of our revised manuscript and Page 56 of our revised supporting information.)

Question 5:

The oxidation peaks of cobalt could influence the assessment of the true OER performance. Please add the reverse sweep LSV curves of samples.

Response:

We greatly appreciate your insightful comments. According to your valuable suggestions, we have added the reverse sweep LSV curves of samples in our revised manuscript.

To make it clear, the related part is listed as follows and highlighted in yellow.

Figure 5. (a) LSV curves and (d) Tafel plots toward ORR in 0.1 M KOH. (b) LSV curves and (e) Tafel plots toward OER, (c) LSV curves and (f) Tafel plots toward HER in 1.0 M KOH for all investigated and commercial catalysts (Pt/C or RuO₂). (g) Open-circuit potential plots, (h) specific capacities, and (i) discharge polarization, charge polarization, and corresponding power density curves for ZABs driven by Co₂-N-HCS-900 or commercial Pt/C+RuO₂. (j) Discharge curves of ZABs at various discharge current densities, and (k) the red light-emitting diode screen powered by two tandem Co₂-N-HCS-900-based ZABs.

(Please find the details in Page 18 of our revised manuscript.)

We really appreciate your valuable and constructive suggestions, and we hope you are satisfied with the present manuscript which is obviously improved with the help from you.

Reviewer #2 (Remarks to the Author):

The author has provided detailed responses to most of the questions, but the scientific significance of this manuscript can't meet the stringent standard of Nature Communications at current stage. Some additional comments are as follows.

Response:

We greatly appreciate your critical comments. Keeping the reviewer's suggestions in mind, we now obtained more evidence to address the proposed questions. Moreover, we have responded point by point to the reviewer's comment as listed in this letter. The changes have been highlighted with yellow in the revised manuscript and supporting information.

Question 1:

There is a lack of positive responses to individual questions such as “Why Co₁₀ is taken as nanoparticles sample for research?” and “The common conclusion of previously reported researches is that the increase of magnetic moment is conducive to the improvement of catalytic activity. However, the relationship between the size of magnetic moment and the level of activity in this work is inconsistent with it. Please make corresponding comparisons and explain accordingly.”

Response:

We greatly appreciate your insightful comments. According to your valuable suggestions, we have investigated the atomization process for the Co₄ and Co₁₆ model in our revised supporting information. It can be seen that the formation of Co_{SA}/C from the decomposition of Co₄ and Co₁₆ nanoparticles requires overcoming a very high kinetic barrier of 3.97 and 3.65 eV, respectively, which is higher than that of the decomposition of Co₁₀ nanoparticle (3.23 eV). Besides, the formation of CoN₄ from Co₁₀ nanoparticles decomposition needs a relatively low kinetic barrier of 0.66 eV to be overcome, which is lower than the kinetic barrier of the formation of CoN₄ from the decomposition of Co₄ (1.50 eV) and Co₁₆ nanoparticles (1.30 eV), respectively. This

implies that the transformation process from the Co₁₀ nanoparticle to single atoms can more easily achieve. Based on the above results and recent literature reports ((1) *Nat. Nanotech.*, **2018**, *13*, 856-861. (2) *Nat. Nanotech.*, **2019**, *14*, 851-857. (3) *J. Am. Chem. Soc.*, **2021**, *143*, 18643-18651. (4) *Angew. Chem. Int. Ed.*, **2023**, 202218630), we take the Co₁₀ nanoparticles as the nanoparticles sample for research.

Indeed, as what you mentioned, we recognize lower spin state is good for reaction, which is inconsistent with the common conclusion of previously reported researches, that is, the increase of magnetic moment is conducive to the improvement of catalytic activity. According to suggestions from you and reviewer 1, we also have added more samples (Co₄ and Co₁₆ nanoparticles models) to establish the brand correlation between spin states and intermediates adsorption/desorption. As shown in Figure 1e and g, it can be clearly captured that the spin magnetic moment presents a positive and quasi-linear correlation ($R^2=0.962$) with the $|\Delta G_{\text{OOH}^*-\text{OH}^*}|$, in the positive sequence from Co₂N₅ dual-atom (0.05 μ_B , 2.55 eV) to CoN₄ single-atom (0.81 μ_B , 3.12 eV) to Co₄ nanoparticles (1.28 μ_B , 3.46 eV) to Co₁₀ nanoparticles (1.48 μ_B , 3.79 eV) and to Co₁₆ nanoparticles (1.87 μ_B , 4.36 eV). As for HER, we can also observe a negative and quasi-linear correlation between the spin magnetic moment and the Gibbs free energy of H* (ΔG_{H^*}), in which the ΔG_{H^*} decreases in a linear manner ($R^2=0.959$) with the upshift of the spin magnetic moment (Figure 1g). The Co₂N₅ dual-atom possesses the highest ΔG_{H^*} of -0.04 eV (close to the ideal value of 0 eV) among the three investigated models, validating that the low spin state enables the Co₂N₅ dual-atom with moderate adsorption and desorption of H* intermediate. Besides, we have added more data to support the viewpoint directly. To disclose the correlation between spin configuration and free energy, the crystal orbital Hamilton population (COHP) is calculated to compare the bonding character of absorbed O* onto Co₁₀, CoN₄ and Co₂N₅ models. As is well known, the positive/negative COHP was on behalf of the bonding/antibonding states, respectively. The Co-O bonding strength can be evaluated by the integrated-COHP (-ICOHP) values, which could quantitatively describe the d-p hybridization strength. It can be seen in Figure S20 that the -ICOHP value was determined to be -0.22 for Co₁₀ model, -0.47 for CoN₄, and -0.45 for Co₂N₅ model, respectively. The middle -ICOHP

value between three investigated models evidently validate the moderated Co-O affinity in the Co_2N_5 model, decoding that it could achieve the optimized O^* adsorption/desorption, thus boosting the ORR/OER activities. As shown in Figure S21, a stronger antibonding state for the Co_2N_5 model appear at Fermi level among three investigated models, suggesting that the more electrons transfer from Co-3d orbital to the unfilled O-2p orbital, thus leading to a lower reaction activation energy and the enhanced conductivity of catalyst with less ohmic loss for better catalytic activity ((1) *Nat. Commun.*, **2021**, 12, 4827. (2) *Angew. Chem. Int. Ed.*, **2020**, 59, 2313).

To make it clear, the related part is listed as follows and highlighted in yellow.

“..... First, the atomization process (Figure 1a) was investigated from the Co_{10} nanoparticles to two kinds of Co single-atom with different coordination of C (named $\text{Co}_{\text{SA}}/\text{C}$) and N atoms (named CoN_4). It can be seen that the formation of $\text{Co}_{\text{SA}}/\text{C}$ from Co_{10} nanoparticle decomposition requires overcoming a very high kinetic barrier of 3.23 eV with a large endothermicity of 3.22 eV, indicating that Co_{10} nanoparticles might be the main form of existence. The formation of CoN_4 from Co_{10} nanoparticles decomposition needs a relatively low kinetic barrier of 0.66 eV to be overcome but with a large exothermicity of 3.84 eV, manifesting that the introduction of N elements can promote the thermal atomization from Co_{10} nanoparticles to CoN_4 single-atom. Besides, the atomization process for the Co_{16} and Co_4 model was also investigated. It can be seen in Figure S1 and S2 that the formation of $\text{Co}_{\text{SA}}/\text{C}$ from the decomposition of Co_4 and Co_{16} nanoparticles requires overcoming a very high kinetic barrier of 3.97 and 3.65 eV, respectively, which is higher than that of the decomposition of Co_{10} nanoparticle (3.23 eV). Besides, the formation of CoN_4 from Co_{10} nanoparticles decomposition needs a relatively low kinetic barrier of 0.66 eV to be overcome, which is lower than the kinetic barrier of the formation of CoN_4 from the decomposition of Co_4 (1.50 eV) and Co_{16} nanoparticles (1.30 eV), respectively. This implies that the transformation process from the Co_{10} nanoparticle to single atoms can more easily achieve.....”

“Next, we investigated the electronic structure of Co_{16} nanoparticles, Co_{10} nanoparticles, Co_4 nanoparticles, CoN_4 single-atom, and Co_2N_5 dual-atom models (Figure S17). The charge density difference and Bader charge results in Figure 1c

validate that the Co atoms of Co₂N₅ dual-atom possess a higher charge of 1.11e than that of CoN₄ single-atom (1.03e) and Co₁₀ nanoparticles (0.11e). As shown in the projected density of states (PDOS) diagram (Figure S18), the electrons of Co-3d orbitals are asymmetrically arranged in the spin channels for Co₁₆ nanoparticles, Co₁₀ nanoparticles, Co₄ nanoparticles, CoN₄ single-atom, and Co₂N₅ dual-atom, showing the magnetic moment of 1.87 μ_B , 1.48 μ_B , 1.28 μ_B , 0.81 μ_B , and 0.05 μ_B , respectively. Such decreased spin magnetic moment mainly results from the redistribution of electrons of Co-3d orbital triggered by the energy levels rearrangement, which would give rise to the increased filling degree of d_{z^2} orbitals and induce the weakened adsorption of reaction intermediates (OOH*, O* and OH* toward ORR/OER, H* toward HER), thus boosting the catalytic activities. Further support for this phenomenon can be demonstrated by the downshift of d band center of Co-3d orbitals from Co₁₆ nanoparticles to Co₁₀ nanoparticles to Co₄ nanoparticles to CoN₄ single-atom, to Co₂N₅ dual-atom, accompanied with the corresponding value of -0.52, -0.71, -1.27, -1.61 and -2.27 eV (Figure 1d).

We also investigated the Gibbs free energy toward ORR/OER/HER of Co₁₀ nanoparticles, CoN₄ single-atom, and Co₂N₅ dual-atom models to afford a deeper understanding of the catalytic mechanism and origin of superior catalytic activities. It is well known that the Gibbs free energy difference between ΔG_{OOH^*} and ΔG_{OH^*} ($|\Delta G_{\text{OOH}^*-\text{OH}^*}|$) could serve as the important ORR/OER reaction descriptor, with an ideal value of 2.46 eV.³⁸⁻⁴⁰ The $|\Delta G_{\text{OOH}^*-\text{OH}^*}|$ is significantly restricted by the adsorption affinity of O*, in which either too strong or too weak adsorption of O* could result in the increased value of $|\Delta G_{\text{OOH}^*-\text{OH}^*}|$.⁴¹ As shown in Figure 1e, g, S18, it can be clearly captured that the spin magnetic moment presents a positive and quasi-linear correlation ($R^2=0.962$) with the $|\Delta G_{\text{OOH}^*-\text{OH}^*}|$, in the positive sequence from Co₂N₅ dual-atom (0.05 μ_B , 2.55 eV) to CoN₄ single-atom (0.81 μ_B , 3.12 eV) to Co₄ nanoparticles (1.28 μ_B , 3.46 eV) to Co₁₀ nanoparticles (1.48 μ_B , 3.79 eV) and to Co₁₆ nanoparticles (1.87 μ_B , 4.36 eV). A universal ΔG_{OOH^*} to ΔG_{OH^*} scaling relation with an average $|\Delta G_{\text{OOH}^*-\text{OH}^*}|$ value of 3.2 eV has been established for most conventional catalysts, whereas the ideal ΔG_{OOH^*} to ΔG_{OH^*} scaling relation possesses an average $|\Delta G_{\text{OOH}^*-\text{OH}^*}|$ value of 2.46 eV

for ideal catalysts (Figure 1f).^{38,39} The ΔG_{OOH^*} to ΔG_{OH^*} coordinate point of Co_2N_5 dual-atom is located at the ideal scaling relations accompanied with the $|\Delta G_{\text{OOH}^*-\text{OH}^*}|$ value of 2.55 eV, which is very close to the 2.46 eV for ideal catalysts toward ORR/OER. This decodes that the energy of O^* intermediates adsorption/desorption for Co_2N_5 is the most appropriate. The possible reason for this phenomenon is that the Co_2N_5 dual-atom with the decreased spin magnetic moment can break the universal ΔG_{OOH^*} to ΔG_{OH^*} scaling relation, thus achieving the ideal balanced O^* adsorption. Such phenomenon can be further supported by the moderate O^* adsorption energy (1.89 eV, Figure S19) of Co_2N_5 dual-atom model among investigated models, decoding that it could achieve the optimized O^* adsorption/desorption, thus boosting the ORR/OER activities. To disclose the correlation between spin configuration and free energy of O^* , the crystal orbital Hamilton population (COHP) is calculated to compare the bonding character of absorbed O^* onto Co_{10} , CoN_4 and Co_2N_5 models. As is well known, the positive/negative COHP was on behalf of the bonding/antibonding states, respectively. The Co-O bonding strength can be evaluated by the integrated-COHP (-ICOHP) values, which could quantitatively describe the d-p hybridization strength. It can be seen in Figure S20 that the -ICOHP value was determined to be -0.22 for Co_{10} model, -0.47 for CoN_4 , and -0.45 for Co_2N_5 model, respectively. The middle -ICOHP value between three investigated models evidently validate the moderated Co-O affinity in the Co_2N_5 model, decoding that it could achieve the optimized O^* adsorption/desorption, thus boosting the ORR/OER activities. As shown in Figure S21, a stronger antibonding state for the Co_2N_5 model appear at Fermi level among three investigated models, suggesting that the more electrons from Co-3d orbital transfer to the unfilled O-2p orbital, thus leading to a lower reaction activation energy and the enhanced conductivity of catalyst with less ohmic loss for better catalytic activity.^{43,44} As for HER, we can also observe a negative and quasi-linear correlation between the spin magnetic moment and the Gibbs free energy of H^* (ΔG_{H^*}), in which the ΔG_{H^*} decreases in a linear manner ($R^2=0.959$) with the upshift of the spin magnetic moment (Figure 1g). The Co_2N_5 dual-atom possesses the highest ΔG_{H^*} of -0.04 eV (close to the ideal value of 0 eV) among the three investigated models, validating that the low spin state enables the Co_2N_5 dual-

atom with moderate adsorption and desorption of H^* intermediate. Besides, the highest H^* adsorption energy of -0.21 eV can be captured among the three investigated models (Figure S22), again validating that it affords moderate adsorption and desorption of H^* intermediate, thus endowing excellent HER activities.”

Figure 1. (a) Calculated relative energies along the stretching pathway of the Co atom from Co₁₀ to Co_{SA}/C or CoN₄ model by CI-NEB, (b) Calculated relative energies of CoN₄, Co₂N₆, and Co₂N₅ models. (c) The charge density difference and Bader charge diagrams, (d) PDOS, (e) linear correlation between magnetic moment and $|\Delta G_{OOH^* \rightarrow OH^*}|$, (f) the ΔG_{OOH^*} to ΔG_{OH^*} scaling for Co₁₆, Co₁₀, Co₄, CoN₄, and Co₂N₅ models relative

to the universal and ideal scaling lines, and (g) linear correlation between the magnetic moment and ΔG_{H^*} . Free energy diagrams of Co-N₄ and Co₂N₅ models for (h) ORR, (i) OER, and (j) HER.

Figure S1. Calculated relative energies along the stretching pathway of the Co atom from Co₄ to Co_{SA}/C or CoN₄ model by CI-NEB.

Figure S2. Calculated relative energies along the stretching pathway of the Co atom from Co₁₆ to Co_{SA}/C or CoN₄ model by CI-NEB.

Figure S17. Optimized atomic configurations of (a) Co_{16} , (b) Co_{10} , (c) Co_4 , (d) CoN_4 and (e) Co_2N_5 model.

Figure S18. PDOS of the Co_{16} , Co_{10} , Co_4 , CoN_4 , and Co_2N_5 models.

Figure S20. Crystal orbital Hamilton population (COHP) analysis corresponding integrated-COHP (-ICOHP) value of the Co₁₀-O (a), CoN₄-O (b) and Co₂N₅-O (c).

Figure S21. The COHP analysis (a) and the magnification around Fermi level image (b) of the Co₁₀-O, CoN₄-O and Co₂N₅-O.

(Please find the details in Page 4-8 of our revised manuscript and Page 1-2 and 22-27 of our revised supporting information.)

Question 2:

In addition, the author mentioned " it was found that the spin state of Co atoms can be harmonized from NP-to-SA-to-DA, in which the Co₂N₅ dual-atom with low spin-state can achieve the ideal balanced adsorption/desorption of O* and H* intermediates." in the article, but no clear evidence was provided in the experiment. EPR is only qualitative analysis and cannot prove the high/low spin states of the metal Co atom.

Response:

We greatly appreciate your insightful comments. According to your valuable suggestions, we have performed the ferromagnetic hysteresis loop and zero-field cooling (ZFC) temperature-dependent magnetic susceptibility measurements to provide more evidence to reveal the detailed information of spin states for the investigated catalysts. As shown in Figure 4g, the ferromagnetic hysteresis loop of investigated catalysts at 300 K exhibits a saturation magnetization. Noted that the saturation

magnetization displays a downward trend from $\text{Co}_{\text{NP}}/\text{HCS-900}$ to $\text{Co}_{\text{SA-N}}/\text{HCS-900}$ to $\text{Co}_2\text{-N-HCS-900}$. An enlarged view of the curve around $H=0$ decodes that the $\text{Co}_2\text{-N-HCS-900}$ affords a lowest coercive magnetic field and residual magnetization (inset of Figure 4g). In order to further reveal the electron spin state of catalysts, the ZFC temperature-dependent magnetic susceptibility was conducted. The unpaired d-band electrons (n) of catalysts can be calculated based on the following formula:

$$\bar{\mu}_{\text{eff}}(t) = \sum_i c_i(t) \mu_i^2 \quad \text{with} \quad \sum_i c_i(t) = 1 \quad (1)$$

$$\mu_{\text{eff}} = \sqrt{n(n+2)} \quad (2)$$

Here, μ_{eff} is the effective magnetic moment of the transition metal ions, c_i is the corresponding concentrations of Co, and n is the number of unpaired d electron of the Co-3d orbitals. As presented in Figure 4h and i, the effective magnetic moment of the $\text{Co}_{\text{SA-N}}/\text{HCS-900}$ and $\text{Co}_2\text{-N-HCS-900}$ can be calculated to be 2.6 and 1.7 μ_{eff} , respectively. The average number of the unpaired electron is 1.0 in Co-3d orbitals for the $\text{Co}_2\text{-N-HCS-900}$, which is lower than that of $\text{Co}_{\text{SA-N}}/\text{HCS-900}$ (1.7), indicating the decreased electron spin polarization from $\text{Co}_{\text{SA-N}}/\text{HCS-900}$ to $\text{Co}_2\text{-N-HCS-900}$. Based on these above-mentioned results, one may conclude that the spin magnetic moment displays a downward trend from the $\text{Co}_{\text{NP}}/\text{HCS-900}$ to $\text{Co}_{\text{SA-N}}/\text{HCS-900}$ to $\text{Co}_2\text{-N-HCS-900}$.

To make it clear, the related part is listed as follows and highlighted in yellow.

“As is well known, the X-band electron paramagnetic resonance (EPR) measurement is a powerful tool to investigate the paramagnetic properties of catalysts. It can be seen in Figure 4f that a strong signal can be detected for the $\text{Co}_{\text{NP}}/\text{HCS-900}$, indicating that it is paramagnetic. A downward trend can be captured from the $\text{Co}_{\text{NP}}/\text{HCS-900}$ to $\text{Co}_{\text{SA-N}}/\text{HCS-900}$ to $\text{Co}_2\text{-N-HCS-900}$, indicating that the spin magnetic moment was decreased from Co nanoparticles to single CoN_4 sites to married Co_2N_5 sites. Notably, no signal appears for the $\text{Co}_2\text{-N-HCS-900}$, possibly due to the formation of the binuclear Co structure with antiferromagnetic coupling sites, again confirming that the married Co_2N_5 structure can be successfully constructed in the $\text{Co}_2\text{-N-HCS-900}$. As shown in Figure 4g, the ferromagnetic hysteresis loop of investigated

catalysts at 300 K exhibits a saturation magnetization. Noted that the saturation magnetization displays a downward trend from $\text{Co}_{\text{NP}}/\text{HCS-900}$ to $\text{Co}_{\text{SA-N}}/\text{HCS-900}$ to $\text{Co}_2\text{-N}/\text{HCS-900}$. An enlarged view of the curve around $H=0$ decodes that the $\text{Co}_2\text{-N}/\text{HCS-900}$ affords a lowest coercive magnetic field and residual magnetization (inset of Figure 4g). In order to further reveal the electron spin configuration of the investigated catalysts, the zero-field cooling (ZFC) temperature-dependent magnetic susceptibility was conducted (Figure 4h and Figure S51). As presented in Figure 4h and i, the effective magnetic moment of the $\text{Co}_{\text{SA-N}}/\text{HCS-900}$ and $\text{Co}_2\text{-N}/\text{HCS-900}$ can be calculated to be 2.6 and 1.7 μ_{eff} , respectively. The average number of the unpaired electron is 1.0 in Co-3d orbitals for the $\text{Co}_2\text{-N}/\text{HCS-900}$, which is lower than that of $\text{Co}_{\text{SA-N}}/\text{HCS-900}$ (1.7), indicating the decreased electron spin polarization from $\text{Co}_{\text{SA-N}}/\text{HCS-900}$ to $\text{Co}_2\text{-N}/\text{HCS-900}$. Based on these above-mentioned results, one may conclude that the spin magnetic moment displays a downward trend from the $\text{Co}_{\text{NP}}/\text{HCS-900}$ to $\text{Co}_{\text{SA-N}}/\text{HCS-900}$ to $\text{Co}_2\text{-N}/\text{HCS-900}$. Ultraviolet photoemission spectroscopy (UPS) was conducted to investigate the electronic state of the $\text{Co}_{\text{NP}}/\text{HCS-900}$, $\text{Co}_{\text{SA-N}}/\text{HCS-900}$ and $\text{Co}_2\text{-N}/\text{HCS-900}$. The work function represents the minimum energy for inner electrons to escape from the surface of catalysts.....”

Figure 4. (a) XANES and (b) comparisons of the q-space magnitudes of Co foil, $\text{Co}_{\text{SA}}\text{-N-HCS-900}$ and $\text{Co}_2\text{-N-HCS-900}$. (c) WT-EXAFS of Co foil, CoO, CoPc, $\text{Co}_{\text{SA}}\text{-N-HCS-900}$, and $\text{Co}_2\text{-N-HCS-900}$ at Co K-edge. (d and e) The corresponding k^3 -weighted EXAFS fitting curves at R space for the $\text{Co}_{\text{SA}}\text{-N-HCS-900}$ and $\text{Co}_2\text{-N-HCS-900}$ (inset: schematic structure of Co-N_4 and Co_2N_5 model, (green: Co, blue: N, red:O)). (f) X-band electron paramagnetic resonance (EPR) spectrum (g) magnetic hysteresis loops at room temperature (300 K) and inset image of the magnification of magnetic hysteresis loops around $H=0$ for the $\text{Co}_{\text{NP}}/\text{HCS-900}$, $\text{Co}_{\text{SA}}\text{-N-HCS-900}$, and $\text{Co}_2\text{-N-HCS-900}$. (h) M-T curves and (i) the corresponding unpaired electron n and effective magnetic moment (μ_{eff}) of the $\text{Co}_{\text{SA}}\text{-N-HCS-900}$ and $\text{Co}_2\text{-N-HCS-900}$. (j) The work function spectrum for the $\text{Co}_{\text{NP}}/\text{HCS-900}$, $\text{Co}_{\text{SA}}\text{-N-HCS-900}$, and $\text{Co}_2\text{-N-HCS-900}$.

Figure S51. M-T curve of the Co_{NP}/HCS-900.

(Please find the details in Page 14-16 of our revised manuscript and Page 56 of our revised supporting information.)

We really appreciate your valuable and constructive suggestions, and we hope you are satisfied with the present manuscript which is obviously improved with the help from you.

REVIEWERS' COMMENTS

Reviewer #1 (Remarks to the Author):

The authors responded the referred problems positively and I have no further questions. The quality of the manuscript improved largely at this time. These experimental and theoretical evidences could justify oneself. The manuscript is recommended to publish in Nature Communications.

Reviewer #2 (Remarks to the Author):

The author has provided a detailed and comprehensive answer to my question.

Responses for the Nature Communications

Manuscript ID: NCOMMS-22-47908C

Title: Establishing libraries for transforming nanoparticles to married dual-atom catalysts

Responses to the comments of reviewers:

Reviewer #1 (Remarks to the Author):

The authors responded the referred problems positively and I have no further questions. The quality of the manuscript improved largely at this time. These experimental and theoretical evidences could justify oneself. The manuscript is recommended to publish in Nature Communications.

Response:

Thank you very much for reviewing our revised manuscript and providing a positive evaluation. The reviewer's comments and suggestions have helped us make our research more comprehensive, rigorous and convincing. Thank you again to the reviewer for the review and valuable opinions.

Reviewer #2 (Remarks to the Author):

The author has provided a detailed and comprehensive answer to my question.

Response:

We extend our sincere gratitude to you for your meticulous evaluation of our revised manuscript, which has yielded a favorable assessment. The insightful comments and suggestions provided by you have significantly enhanced the comprehensiveness, rigor, and persuasiveness of our research. We express our heartfelt appreciation to the reviewer for the reviewer invaluable critique and constructive insights.